# ShapeLLM-Omni: A Native Multimodal LLM for 3D Generation and Understanding

**Junliang Ye**[1,3*]  **Zhengyi Wang**[1,3*]  **Ruowen Zhao**[1*]  **Shenghao Xie**[2]  **Jun Zhu**[1,3†]

Tsinghua University[1]   Peking University[2]   ShengShu [3]

`https://github.com/JAMESYJL/ShapeLLM-Omni/`

## Abstract

Recently, the powerful text-to-image capabilities of GPT-4o have led to growing appreciation for native multimodal large language models. However, its multimodal capabilities remain confined to images and text. Yet beyond images, the ability to understand and generate 3D content is equally crucial. To address this gap, we propose ShapeLLM-Omni—a native 3D large language model capable of understanding and generating 3D assets and text in any sequence. First, we train a 3D vector-quantized variational autoencoder (VQVAE), which maps 3D objects into a discrete latent space to achieve efficient and accurate shape representation and reconstruction. Building upon the 3D-aware discrete tokens, we innovatively construct a large-scale continuous training dataset named 3D-Alpaca, encompassing generation, comprehension, and editing, thus providing rich resources for future research and training. Finally, we perform instruction-based fine-tuning of the Qwen-2.5-vl-7B-Instruct model on the 3D-Alpaca dataset, equipping it with native 3D understanding and generation capabilities. Our work represents an effective step toward extending multimodal large language models with fundamental 3D intelligence, paving the way for future advances in 3D-native AI.

## 1   Introduction

Large language models have made significant achievements, including text-only language models (LLMs) Achiam et al. [2023], Liu et al. [2024a], Bai et al. [2023], Touvron et al. [2023], Multimodal Large Language Models (MLLMs) that can understand images Hurst et al. [2024a], GLM et al. [2024], Team [2024], video Guo et al. [2025], Cheng et al. [2024], Maaz et al. [2023], Li et al. [2024b] and 3D Wang et al. [2024b], Siddiqui et al. [2024a], Chen et al. [2023a, 2025b] content. These models employ similar transformer architectures, using dedicated encoders to model each modality independently, thereby integrating images, video, and 3D modalities into existing LLMs.

Recently, ChatGPT-4o Hurst et al. [2024a] has demonstrated remarkable performance. By natively incorporating image generation and understanding into the large language model (LLM) architecture, it enables more fine-grained and precise control through human instructions. However, its multimodal capabilities remain confined to images and text, limiting its potential in more complex spatial domains.

In this work, we propose a unified approach to integrate 3D generation and understanding into a pre-trained multimodal large language model (MLLM). Enhancing LLMs with native 3D capabilities is crucial for downstream applications such as 3D content creation, robotics, digital twins, and immersive virtual environments.

Our method adopts a fully next-token prediction paradigm, which ensures natural compatibility with joint training and large-scale scalability. We leverage a VQVAE to encode 3D meshes into compact

---

*Equal contribution

†Corresponding author.

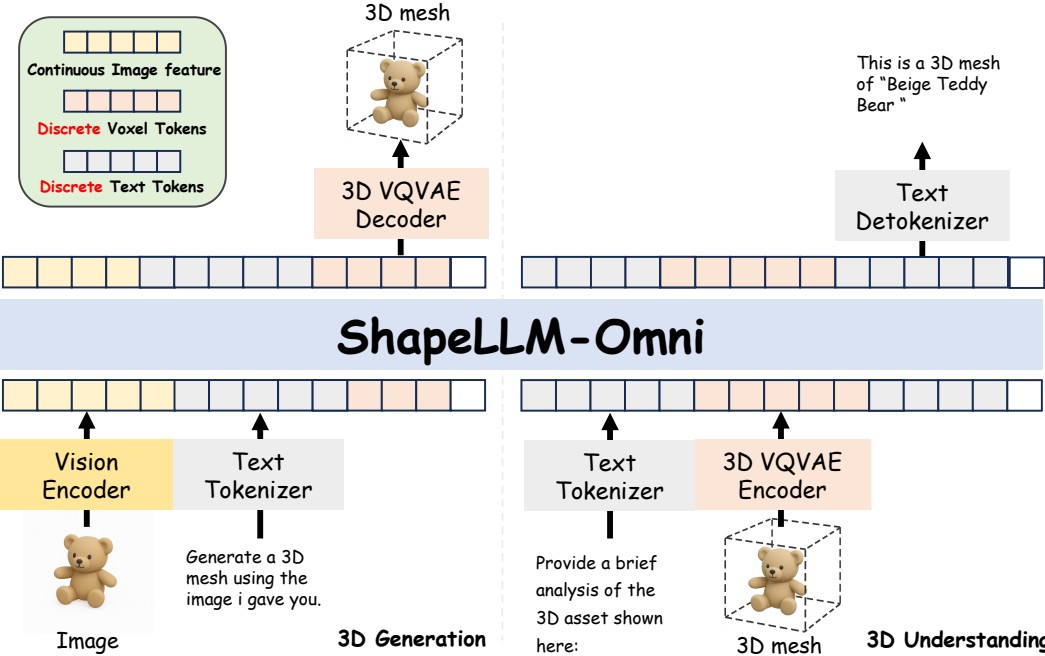

Figure 1: ShapeLLM-Omni inherits Qwen2.5-vl's strong multimodal capabilities and additionally supports text-to-3D, image-to-3D, 3D captioning, and 3D editing using text instruction.

discrete tokens, enabling a unified representation. These tokens are utilized for both understanding and generating 3D meshes, following a format analogous to language modeling.

To enable LLMs with 3D ability, we construct a comprehensive training dataset using 3D shapes from a mixture of 3D datasets Deitke et al. [2023a,b], Collins et al. [2022], Chang et al. [2015]. We construct interleaved 710k text/image-3D pairs to enable the model for basic 3D understanding ability and text/image to 3D generation ability.

Furthermore, to enable interactive 3D mesh editing, we introduce a novel dataset of 62k paired 3D meshes and corresponding text-based editing instructions. This facilitates fine-grained manipulation of 3D assets through natural language, making real-time editing more intuitive and controllable.

After that, we train an LLM on the corpus. We resume from Qwen-2.5-VL-Instruct-7B Bai et al. [2025] to utilize the effective of its large-scale pre-training on text and images. Our model demonstrates a wide range of capabilities, including: (1) generating 3D content from language instructions; (2) generating 3D objects from image inputs; (3) interactively editing 3D assets using natural language; (4) understanding and interpreting 3D meshes for semantic and geometric reasoning.

In all, our contributions are:

- We propose a novel framework for unified 3D object generation and understanding based on a fully autoregressive next-token prediction paradigm.
- We present the 3D-Alpaca dataset for training large language models (LLMs) with 3D capabilities. Comprising 3.46 billion tokens, it covers three core tasks: 3D generation, 3D understanding, and 3D editing.
- Our experimental results provide strong empirical evidence supporting the effectiveness of the proposed method.

## 2 Related Work

### 2.1 3D Mesh Generation

The remarkable achievement of 2D diffusion models Ho et al. [2020], Rombach et al. [2022] has facilitated the exploration of 3D generative models. Early 3D generation methods Poole et al. [2022],

Wang et al. [2023e], Chen et al. [2023b], Lin et al. [2023], Raj et al. [2023], Li et al. [2023b], Sun et al. [2023], Chen et al. [2024h], Wang et al. [2022], Tang et al. [2023], Yi et al. [2024] often rely on SDS-based optimization to distill 3D content due to the limited 3D data, but encounter challenges such as long optimization time and Janus problem. Subsequent works such as Wang and Shi [2023], Shi et al. [2023b], Wang et al. [2023c], Liu et al. [2025a], Ye et al. [2024b], Qiu et al. [2024], Chen et al. [2024a] enhance semantic consistency across different views during multi-view image synthesis. To minimize generation time, more recent approaches Long et al. [2024], Zhao et al. [2024], Liu et al. [2023d,c], Shi et al. [2023a], Weng et al. [2023], Liu et al. [2023b], Wu et al. [2024a], Chen et al. [2024i], Voleti et al. [2024], Ye et al. [2024a], Liu et al. [2024b] adopt a two-stage pipeline that integrates multi-view image prediction with 3D reconstruction to produce 3D models. LRM Hong et al. [2023a] and other works Tang et al. [2024a], Wei et al. [2024], Ziwen et al. [2024], Li et al. [2023a], Xu et al. [2023], Wang et al. [2023a], Siddiqui et al. [2024b], Zhang et al. [2024a,b], Zou et al. [2024], Xu et al. [2024a], Nawrot et al. [2021], Wang et al. [2024c] build on a feed-forward reconstruction model and predict 3D structures within seconds. Additionally, native 3D diffusion models Zhao et al. [2023], Wang et al. [2023b], Wu et al. [2024b], Yang et al. [2024], Huang et al. [2025b], Yang et al. [2024], Zhang et al. [2024c], Xiang et al. [2024], Chen et al. [2024f], Li et al. [2024a], Wu et al. [2024b], Ye et al. [2025] encode 3D objects into a VAE latent and adapt a latent diffusion model on the resulting representations for comprehensive 3D understanding. Nevertheless, the above methods treat 3D objects as numerical fields Mildenhall et al. [2021], Kerbl et al. [2023] and extract meshes using Marching Cubes Lorensen and Cline [1998], which are not easily represented as discrete tokens.

## 2.2   Autoregressive 3D Generation

Inspired by the success of auto-regressive models in language and image synthesis, some pioneering works Siddiqui et al. [2024a], Chen et al. [2024d], Weng et al. [2024a] have explored their use in 3D shape generation. They adopt VQVAE Van Den Oord et al. [2017] to compress 3D shapes into latent spaces, which are subsequently quantized into discrete tokens for learning via an auto-regressive transformer. Instead of employing VQVAE, other studies Chen et al. [2024e, 2025a], Liu et al. [2025d,b,c], Yang et al. [2025], Weng et al. [2024b], Tang et al. [2024b], Hao et al. [2024], Zhao et al. [2025] have proposed specialized mesh tokenization techniques that transform mesh vertices and faces into compact discrete token sequences, while preserving the original complex geometric details. These approaches enable the auto-regressive model to effectively generate meshes in a face-by-face manner. Building on 3D auto-regressive models, LLaMA-Mesh Wang et al. [2024b] explores the integration of natural language instructions with mesh generation and understanding, enabling interactive 3D content creation through a unified framework. However, it treats the 3D OBJ mesh file as text for language model to process, which overlooks the inherent topological structures of 3D data.

## 2.3   Unified Models for Multimodal Understanding and Generation

Extending large language models (LLMs) to process, generate, and comprehend multiple modalities—such as vision and language—within a unified framework has become a major research frontier. Previous studies Bai et al. [2023], Chen et al. [2024g], Alayrac et al. [2022] have advanced this direction by equipping LLMs with visual understanding capabilities for multimodal tasks. Concurrently, other works Team [2024], Liu et al. [2024c], Wang et al. [2024a], Xie et al. [2024], Zhou et al. [2024] have proposed the integration of image and text generation through specialized visual tokenizers. More recently, ChatGPT-4o has further propelled this progress, achieving state-of-the-art performance in both visual comprehension and image synthesis. Beyond 2D modalities, a growing body of research Hong et al. [2023b], Xu et al. [2024b], Qi et al. [2024a], Xue et al. [2023], Huang et al. [2025a], Chen et al. [2024b], Huang et al. [2024], Kang et al. [2025], Chen et al. [2024c], Wang et al. [2023d] has extended LLMs to 3D content understanding, primarily through point cloud representations. However, point clouds often lack fine-grained geometric detail and are challenging to acquire in real-world settings, limiting their applicability for interactive generation. Despite these advancements, there remains a notable gap: very few models are capable of jointly processing and generating text, images, and 3D data in an integrated manner. To bridge this gap, we introduce a 3D VQVAE module that encodes 3D shapes into discrete representations, enabling autoregressive models to perform unified multimodal understanding and generation across text, images, and 3D content.

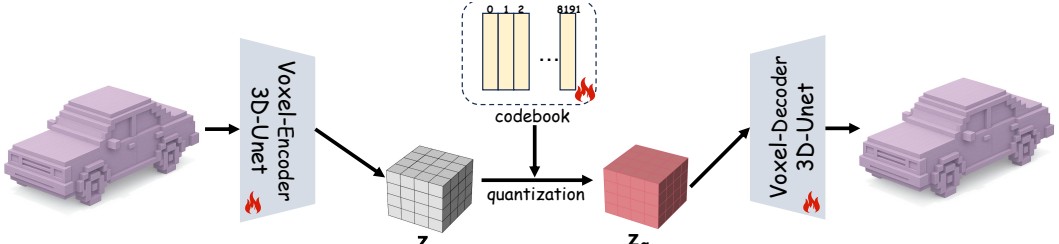

Figure 2: The pipeline of 3D VQVAE, which can compress voxels into discrete tokens.

# 3  Method

Table 1: **Modality comparison**. In contrast to the task-specific model architectures of SAR3D and Trellis, ShapeLLM-Omni achieves cross-modal alignment by jointly modeling text and 3D representations in a shared latent space, enabling unified understanding and generation capabilities.

|  | Input Modality | | | | Output Modality | | |
|---|---|---|---|---|---|---|---|
|  | Text | Image | 3D | Unified model | Text | Image | 3D |
| SAR3D Chen et al. [2025b] | ✓ | ✓ | ✓ |  | ✓ |  | ✓ |
| Trellis Xiang et al. [2024] | ✓ | ✓ |  |  |  |  | ✓ |
| PointLLM Xu et al. [2024b] | ✓ |  | ✓ | ✓ | ✓ |  |  |
| LLaMA-Mesh Wang et al. [2024b] | ✓ |  | ✓ | ✓ | ✓ |  | ✓ |
| ChatGPT-4o Hurst et al. [2024b] | ✓ | ✓ |  | ✓ | ✓ | ✓ |  |
| Qwen-2.5vl Bai et al. [2025] | ✓ | ✓ |  | ✓ | ✓ |  |  |
| **ShapeLLM-Omni (ours)** | ✓ | ✓ | ✓ | ✓ | ✓ |  | ✓ |

## 3.1  Overview

Figure 1 provides an overview of our native Multimodal LLM framework, which can handle mixed sequences of text, images, and 3D data and produce corresponding text or 3D outputs. We begin by converting 3D assets into discrete tokens using a 3D VQVAE (Sec. 3.3), which allows us to leverage the same transformer architecture for both 3D and text token sequences. Subsequently, we assemble a comprehensive 3D supervised fine-tuning dataset, 3D-Alpaca (Sec. 3.4), covering text-to-3D generation, image-to-3D generation, 3D captioning, and 3D editing.

## 3.2  Architecture

As shown in Figure 1, we represent both text and 3D data as sequences of discrete tokens, enabling fully autoregressive multimodal generation. This design allows for flexible input and output across modalities in any order. While we adopt token-based representations for both text and 3D modalities, we use continuous features for images. This is because images are only involved in understanding tasks, whereas 3D data supports both understanding and generation. Such a unified modeling approach—based on early fusion—facilitates better modality integration within the language model. Compared to prior work in the 3D domain Table 1, our model is the first unified auto-regressive framework that supports text-to-3D, image-to-3D, 3D understanding, and 3D editing in a single system. It also marks the first attempt at a ChatGPT-4o-style model tailored for 3D tasks.

## 3.3  3D VQVAE

In this section, we introduce our 3D representation—voxels—explain why we chose voxels, and how we compress voxels into discrete tokens using a 3D VQVAE. Finally, we describe how to reconstruct high-quality 3D meshes from voxels.

**Voxel-Based Representation**  3D assets can be represented in various ways—such as voxels, vecset Zhang et al. [2023], Face-Vertex representation Wang et al. [2024b], Point Clouds Xu et al. [2024b], or Gaussian splats Kerbl et al. [2023]. In this work, we adopt low-resolution voxels as

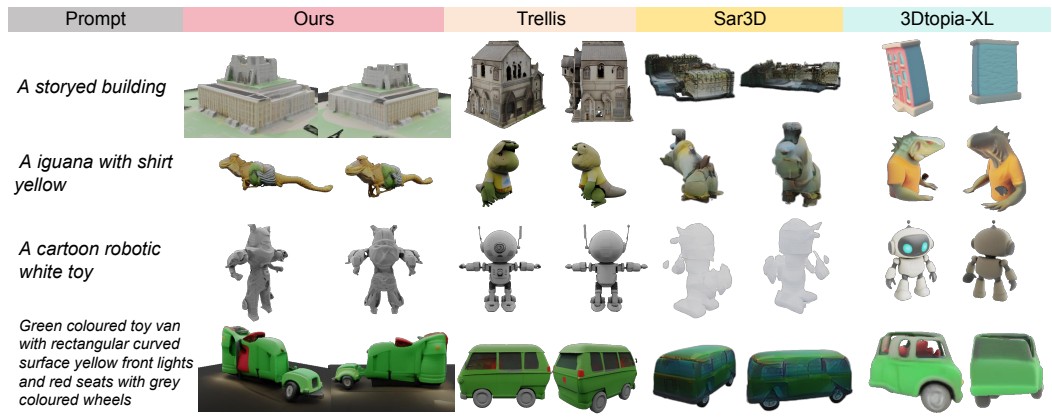

| Prompt | Ours | Trellis | Sar3D | 3Dtopia-XL |
|--------|------|---------|-------|-----------|
| *A storyed building* | | | | |
| *A iguana with shirt yellow* | | | | |
| *A cartoon robotic white toy* | | | | |
| *Green coloured toy van with rectangular curved surface yellow front lights and red seats with grey coloured wheels* | | | | |

Figure 3: Our 3D-Alpaca dataset comprises 3D generation, understanding, and editing components, providing a comprehensive foundation for training and evaluating 3D large language models.

our 3D representation. We now explain the rationale behind this choice. First, we do not adopt the Face-Vertex representation because it quantizes mesh geometry into discrete spatial tokens, resulting in excessively long token sequences that hinder the training efficiency of unified models. Second, we do not use VecSets-based representations. On one hand, VecSets encode highly informative and continuous geometric features, making it challenging to train a complete 3D VQ-VAE for their encoding. On the other hand, VecSets are inherently implicit, whereas voxels provide explicit and structured spatial representations that are more suitable for 3D editing tasks requiring direct geometric manipulation. In contrast, voxels strike a favorable balance between compactness and expressiveness: they compress complex 3D information into a much smaller latent space, facilitating efficient training, while effectively preserving an asset's essential shape and skeletal structure, thereby providing sufficient geometric cues for language models. Moreover, open-source reconstruction models can be readily leveraged to convert coarse-resolution voxels into high-quality, detail-rich meshes.

**Model Architecture** We adopt a $64^3$ voxel grid resolution, as voxels at this resolution strike the optimal balance for modeling 3D skeletons, preserving essential structural details while avoiding excessive redundancy Xiang et al. [2024]. Although voxel representations are compact, even modeling a single 3D object with a $64^3$ voxel grid still requires $64^3$ tokens—far beyond what a large language model can handle. Therefore, we further compress voxels using a 3D VQVAE Xiang et al. [2024]: first, we encode the $64^3$ grid into a $16^3$ latent grid; then we serialize it into 4096 tokens. However, 4096 tokens remain too long. Inspired by Team [2024], which represents images as 1024 tokens, we concatenate every four neighboring tokens along the channel dimension—transforming the original 4096 tokens with 8 channels into 1024 tokens with 32 channels. Finally, we employ an 8192-entry codebook to compress the voxels into 1024 discrete tokens. In all, we represent a single 3D object using 1024 discrete tokens, for both generation and understanding.

**Shape Reconstruction** Although we employ voxel-based representations for 3D shape generation, practical deployment often necessitates converting voxels into meshes for downstream applications. To address this, we adopt the approach proposed by Xiang et al. Xiang et al. [2024], which utilizes a Rectified Flow model to refine and complete voxel information, enabling high-quality mesh reconstruction. By first generating 3D shapes in the voxel domain and then converting them into meshes using this method, our framework achieves a balance between precision and efficiency. This hybrid representation allows large language models to exert fine-grained control over 3D content generation while avoiding the computational burden associated with high-resolution geometry.

### 3.4 3D-Alpaca Dataset Construction

Although a wealth of datasets has been developed for the supervised fine-tuning of multimodal large-language models, dialogue data within the 3D LLM Hong et al. [2023b], Chen et al. [2025b], Xu et al. [2024b] domain remains relatively scarce. To bridge this gap, we introduce 3D-alpaca, a comprehensive dataset encompassing tasks in 3D content generation, comprehension, and editing.

**3D Generation and Understanding Dataset**  We select a high-quality subset of approximately 712k 3D assets from Trellis Xiang et al. [2024] and internal collection. For the image collection, each asset is rendered into a 2D image, and a random offset is applied to the frontal view to create the input. Moreover, these rendered images also underpin the construction of the editing dataset in the Sec. 3.4. To generate the text collection and enable early fusion across all three modalities, we render four orthogonal views—front, back, left, and right—of each asset. These multi-view images are then input into the base model Qwen-2.5-VL-Instruct Bai et al. [2025] to generate descriptive captions. The resulting captions are utilized both as prompts for text-to-3D generation and as ground-truth targets for 3D-to-text captioning tasks.

**3D Edited Dataset**  We aim to build a 3D asset-editing dataset composed of paired 3D assets, where each pair is linked to a specific editing instruction. Despite recent advances in 3D content creation, the field still lacks a model capable of performing consistent edits on 3D assets. In light of the promising performance of current image-editing models, we therefore adopt an image-mediated pipeline: first rendering each 3D asset into images and applying an image-editing model, then reconstructing the edited images back into 3D assets via an image-to-3D generation method. Based on the multimodal alignment demonstrated and with the aim of equipping the model with ChatGPT-4o–level editing capabilities, we follow a six-step pipeline.

(1) *Category*: We reference the data distribution of Objaverse-XL Deitke et al. [2023a] and manually selected the 100 most representative and frequent object categories, such as cars, tables, cabinets, human figures, etc.

(2) *Asset Classification*: Using ChatGPT-4o, we classify the 3D assets in our dataset into fine-grained subcategories, with the frontal view renderings of each asset as input. From the 3D asset dataset, we filtered 311k assets belonging to the predefined 100 major categories.

(3) *Editing-Prompt Definition*: We provide the category names to ChatGPT-4o and instruct it to generate 20 feasible editing-prompts for each category. The instruction given to ChatGPT-4o is: "For each given category name, suggest potential image editing operations that could be applied to objects of that category." Next, we manually review each generated editing prompt and retain only those that meet both our technical feasibility and visual engagement criteria, resulting in 371 unique editing prompts (e.g: "Replace the chair's backrest with a mesh frame").

(4) *Asset Sampling & Annotation*: Due to time and resource constraints, we build a compact, high-quality dataset of editing prompts rather than applying every possible editing prompt to each asset. Specifically, we allocate 200 assets to each editing prompt.

(5) *Editing-Image Pair Collection*: For each sampled asset, we provide ChatGPT-4o with its frontal render plus the chosen editing-prompt, and ChatGPT-4o produces the corresponding edited image, yielding image-level editing pairs. After filtering out erroneous cases, we end up with 70k valid editing samples.

(6) *3D reconstruction*: Finally, we employ Trellis Xiang et al. [2024] to convert the curated images into 3D assets, resulting in 3D pairs before/after editing.

**Dialogue Data Construction**  We define 25 dialogue templates per task (e.g., "Generate a 3D asset of prompt/images") and encode all 3D assets into discrete token sequences with our pre-trained 3D VQVAE (Sec. 3.3). For each 3D-edit instance, we randomly select 6 templates from a pool of 25; for all other instances, we randomly assign one template each. By merging the tokens with these templates, we create a training corpus of 2.5 million 3D dialogues.

**General Conversation**  To ensure the model's general conversational capability, we adopt Ultra-Chat Ding et al. [2023] as our text-only dataset, with its data distribution shown in the Table 2. For additional details, please refer to the Appendix.

**Putting these together**  After data processing and construction, we finally arrive at the 3D-Alpaca dataset. As shown in the Table 2, the dataset includes four types of tasks: image-to-3D, text-to-3D, 3D-to-caption, and 3D-editing. Together, these four subsets form a total of 2.56 million samples, comprising 3.46 billion tokens. To ensure the large language model retains its original reasoning and dialogue capabilities, we additionally include the UltraChat Ding et al. [2023] dataset, a high-quality, large-scale multi-turn dialogue corpus.

Table 2: **Corpus Data Proportions** An overview of token and item counts in the training corpus, covering two datasets: the *3D-Alpaca* dataset, which includes four task types—Text-to-3D, Image-to-3D, 3D-to-Caption, and 3D-Editing—and the text-only *UltraChat* dataset Ding et al. [2023]

|  | Text-To-3D | Image-To-3D | 3D-to-Caption | 3D-Edit | 3D-All | Text-Only |
|---|---|---|---|---|---|---|
| Token count | 0.77B | 1.01B | 0.77B | 0.91B | 3.46B | 2.16B |
| Item count | 712k | 712k | 712k | 420k | 2.56M | 1.47M |

## 4 Experiments

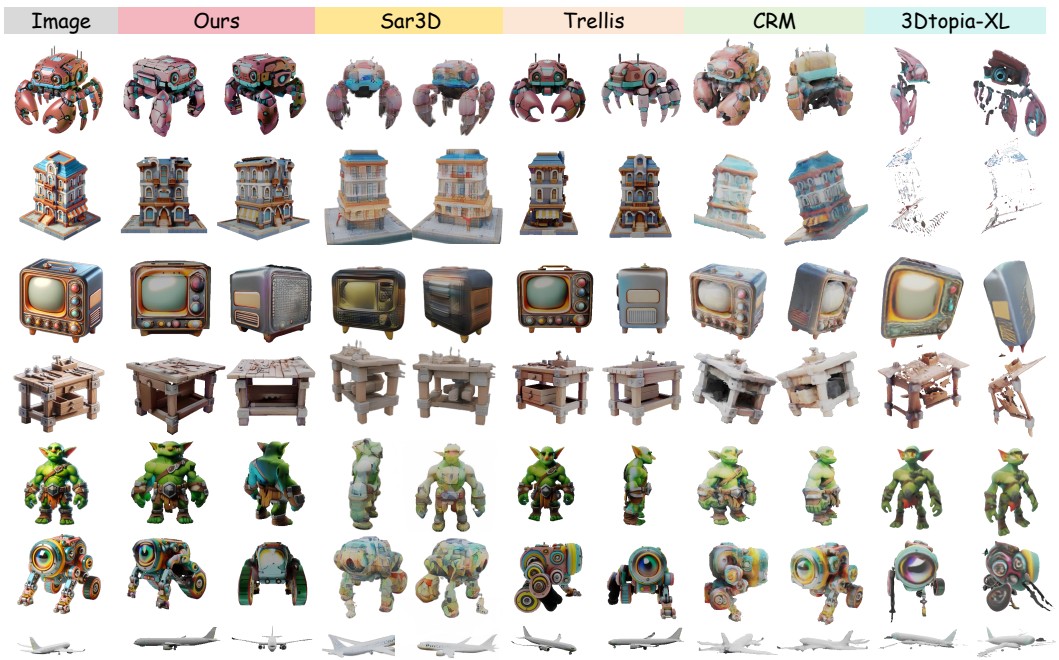

Figure 4: **Comparisons with other baselines on the image-to-3D task.** Our results demonstrate more complete geometry and high-fidelity textures compared to baselines, enabling photorealistic image-to-3D generation.

### 4.1 Implementation Details

For training our 3D VQVAE, we adopt a 3D U-Net VAE architecture introduced in Trellis Xiang et al. [2024]. Our training follows a two-stage strategy: In Stage 1, we freeze the VAE's pre-trained parameters and train only the codebook. In Stage 2, we unfreeze the VAE and jointly fine-tune it with the codebook. Concretely, each stage runs for 1000 steps on 48 NVIDIA H100 GPUs with a batch size of 25, while the learning rate decays from $5 \times 10^{-3}$ to $5 \times 10^{-5}$. For the training of ShapeLLM-Omni, we use Qwen-2.5-VL-Instruct-7B Bai et al. [2025], a multimodal large language model (MLLM) with image-understanding capability, as our backbone. Specifically, we extend its base architecture by adding the 8192 3D VQVAE codebook. To preserve its original image-understanding skills, we freeze the parameters of Qwen2.5-vl's visual encoder. While training, the learning rate decays from $5 \times 10^{-5}$ to $5 \times 10^{-6}$, with a per-GPU batch size of 2 and gradient accumulation over 2 steps. The model is trained for 15 epochs on 48 NVIDIA H100 GPUs.

### 4.2 Quantitative comparisons

**Language and Conversational Abilities**    Table 3 presents quantitative results evaluating language abilities. The table provides a comparison with models: LLaMA-Mesh Wang et al. [2024b], Chameleon Team [2024], and Qwen2.5-vl Bai et al. [2025]. The metrics include SIQA Sap et al.

[2019], PIQA Bisk et al. [2020], MMLU Hendrycks et al. [2020], and GSM8K Cobbe et al. [2021]. Fine-tuned on 3D-Alpaca for both 3D mesh generation and comprehension, our ShapeLLM-Omni maintains language understanding and reasoning performance on par with baseline models. The result demonstrates that ShapeLLM-Omni effectively extends the MLLM's capabilities to 3D content generation while preserving its native language capabilities.

Table 3: **Language capabilities comparison**. We provide a comparison with models: LLaMA-Mesh Wang et al. [2024b], Chameleon Team [2024], and Qwen2.5-vl Bai et al. [2025]. The metrics include SIQA Sap et al. [2019], PIQA Bisk et al. [2020], MMLU Hendrycks et al. [2020], and GSM8K Cobbe et al. [2021]. Fine-tuned on 3D-Alpaca for both 3D mesh generation and comprehension, our ShapeLLM-Omni maintains language understanding and reasoning performance. The table highlights the optimal values in bold and the suboptimal values with underlining.

| Metric | Qwen2.5-vl-7B | ShapeLLM-Omni-7B | Chameleon-7B | LLaMA-Mesh-8B |
|--------|---------------|------------------|--------------|---------------|
| MMLU   | **66.9**      | 63.9             | 59.4         | 57.4          |
| PIQA   | **81.0**      | 78.6             | 79.6         | 78.9          |
| GSM8K  | 42.9          | 55.1             | **66.9**     | 33.1          |
| SIQA   | 40.7          | 41.0             | **57**       | 40.4          |

**3D VQVAE Reconstruction Evaluation**  To assess the reconstruction quality of our 3D VQVAE model, we randomly select 1000 samples from the test set and feed them into the model. We then calculate several metrics between original and reconstructed voxel grids, including IoU, Recall, Precision, F1 and Chamfer Distance. These results, summarized in the table 4, indicate that our 3D VQVAE model preserves geometric structure with high fidelity, providing a reliable reconstruction basis for following generation tasks.

Table 4: **Quantitative Evaluation of 3D VQVAE reconstruction performance.** We report IoU, Recall, Precision, F1, and Chamfer Distance between original and reconstructed voxel grids. The results demonstrate that our 3D VQVAE effectively preserves geometric structure with high fidelity.

|          | IOU    | Average Recall | Average F1 | Average Precision | Chamfer Distance |
|----------|--------|----------------|------------|-------------------|------------------|
| 3D VQVAE | 0.9168 | 0.9357         | 0.9450     | 0.9549            | 0.0214           |

**3D Generation**  We compare our methods on both text-to-3D and image-to-3D generation tasks against CRM Wang et al. [2024c], SAR3D Chen et al. [2025b], 3DTopia-XL Chen et al. [2024f], and TRELLIS Xiang et al. [2024]. When evaluating the generation performance of ShapeLLM-Omni, we set the model's top-k parameter equal to the size of the 3D vocabulary (8192), with top-p=0.7 and temperature=0.7. Regarding the dialogue templates, the image-to-3D template is formulated as: *"Create a 3D asset using the following image: <image>"*, while the text-to-3D template is expressed as: *"Please generate a 3D mesh based on the prompt I provided: <prompt>"*. Quantitative evaluations are conducted using image and text prompts sampled from the Toys4K Stojanov et al. [2021] test dataset, with the results summarized in Table 5. To assess the overall quality of the generated 3D outputs, following Xiang et al. [2024], we compute Frechet Distance (FD) Heusel et al. [2017] and Kernel Distance (KD) Bińkowski et al. [2018] using Inception-V3 Szegedy et al. [2016] features. Additionally, we report the CLIP score Radford et al. [2021] to measure the semantic alignment between the generated outputs and their input prompts. As shown in the Table 5, our generation results outperform all baseline methods except for Trellis.

**3D Understanding**  Following the evaluation settings provided by PointLLM Xu et al. [2024b], we test the same metrics on the benchmark dataset used by PointLLM. We adopt the same curated test set to assess the 3D-to-caption task. The dialogue prompt is structured as: *"<mesh>. Caption this 3D model in detail."*. As shown in Table 6, our ShapeLLM-Omni demonstrates strong 3D understanding capabilities, with performance second only to PointLLM, which is specifically tailored for single-task 3D understanding.

Table 5: Comparison of methods on Text-to-3D and Image-to-3D tasks. We scale KD by ($\times 10^2$).

| Method | Text-to-3D | | | Image-to-3D | | |
|---|---|---|---|---|---|---|
| | CLIP↑ | $FD_{incep}$ ↓ | $KD_{incep}$ ↓ | CLIP↑ | $FD_{incep}$ ↓ | $KD_{incep}$ ↓ |
| CRM | - | - | - | 76.1 | 14.7 | 0.12 |
| 3DTopia-XL | - | - | - | 76.5 | 49.5 | 1.63 |
| SAR3D | 23.9 | 27.2 | 0.28 | 84.70 | 20.6 | 0.17 |
| Trellis | **30.8** | **18.3** | **0.19** | **85.0** | **8.31** | **0.07** |
| **ShapeLLM-Omni (ours)** | 26.7 | 25.9 | 0.25 | 84.5 | 12.2 | 0.09 |

Table 6: **3D object captioning results Xu et al. [2024b] on Objaverse Deitke et al. [2023a]**. As can be seen from the table, our model achieves better performance on 3D understanding/caption tasks. "*" indicate PointLLM was prompted for shorter captions with no more than 20 words.

| Model | B-1 | R-L | METEOR | S-BERT | S-CSE |
|---|---|---|---|---|---|
| InstructBLIP-13B Wenliang et al. [2023] | 4.65 | 8.85 | 13.23 | 45.90 | 48.86 |
| LLaVA-13B Liu et al. [2023a] | 4.02 | 8.15 | 12.58 | 46.37 | 45.90 |
| GPT4Point Qi et al. [2024b] | 8.45 | 10.11 | 13.13 | 40.31 | 42.88 |
| ShapeLLM Qi et al. [2024a] | 17.88 | 19.24 | 17.96 | 48.52 | 49.98 |
| 3D-LLM Hong et al. [2023b] | 16.91 | 19.48 | 19.73 | 44.48 | 43.68 |
| PointLLM-13B Xu et al. [2024b] | 3.38 | 7.23 | 12.26 | 47.91 | 49.12 |
| PointLLM-13B* Xu et al. [2024b] | 17.09 | 20.99 | 16.45 | **50.15** | **50.83** |
| **ShapeLLM-Omni (ours)** | **18.51** | **21.37** | **19.89** | 49.34 | 50.72 |

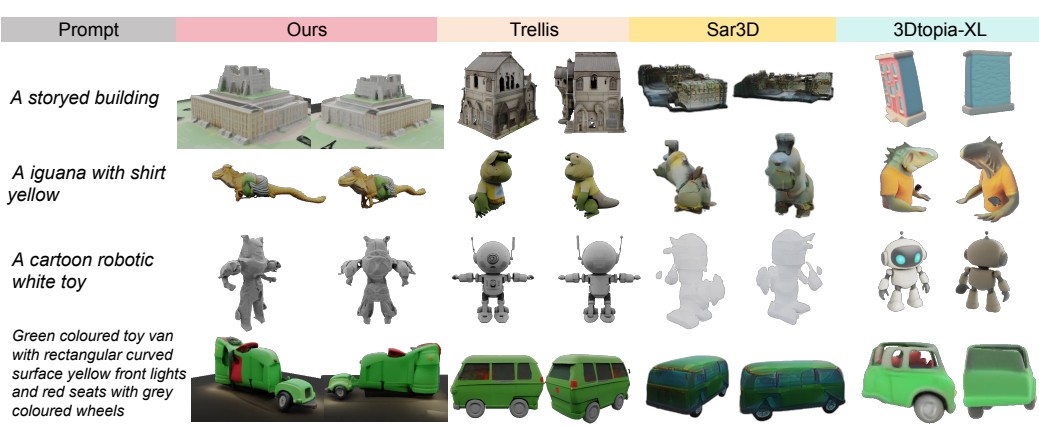

Figure 5: **Comparisons with other baselines on text-to-3d task.** Our method achieves better text alignment, with 3D shapes accurately reflecting input descriptions.

## 4.3 Qualitative comparisons

**3D Generation** To evaluate the effectiveness of our image-conditioned generation, we compare against baselines including SAR3D, TRELLIS, CRM, and 3Dtopia-XL. As illustrated in Figure 4, the baselines exhibit limitations in capturing fine-grained visual features, suffering from geometric distortions and texture misalignments. In contrast, our method generates high-quality 3D meshes that preserve both geometry and appearance details. Moreover, our generation quality matches that of TRELLIS, our base model and performance upper bound, due to the integration of a well-trained 3D VQVAE and a carefully constructed image-to-3D dataset for LLM fine-tuning. For text-to-3D tasks, Figure 5 presents qualitative comparisons among baselines. The input prompts are randomly generated by ChatGPT-4o to cover a diverse range of objects. Since 3Dtopia-XL does not support text-to-3D tasks, we use ChatGPT-4o to generate reference images from the prompts. These images are then used as input for image-to-3D generation. It is evident that our method achieves precise alignment with the text prompts and excels at generating intricate, coherent details.

**3D Editing**   As shown in Figure 6, ShapeLLM-Omni can edit 3D assets according to user-provided instructions while maintaining good identity consistency.

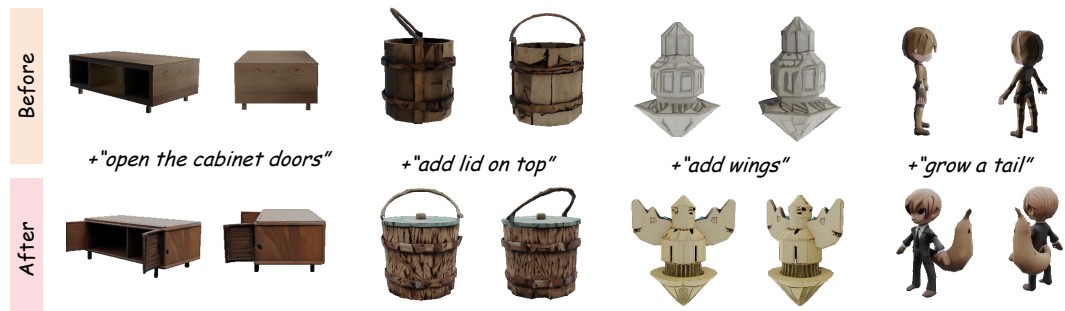

Figure 6: **Some cases of 3D editing result from our method.** Our method enables the editing of 3D assets based on textual instructions while preserving their original identity and visual consistency.

## 4.4   Compared with Trellis

Table 7: Comparison of Trellis and ShapeLLM-Omni in text-to-3D generation performance after task-specific fine-tuning.

| Model | CLIP↑ | $FD_{incep} \downarrow$ | $KD_{incep} \downarrow$ |
|---|---|---|---|
| Trellis | **30.8** | **18.3** | **0.19** |
| ShapeLLM-Omni | 26.7 | 25.9 | 0.25 |
| ShapeLLM-Omni (Overfitting) | 30.1 | 18.9 | 0.21 |

Our results are slightly inferior to Trellis due to two main factors.

1) Trellis employs separate models for text-to-3D and image-to-3D generation, whereas our ShapeLLM-Omni unifies six tasks—text-to-3D and image-to-3D generation, 3D understanding, 3D editing, image understanding, and text reasoning—within a single model that also supports interactive conversation. This all-in-one design introduces optimization trade-offs that can affect generation quality. To verify this, we fine-tuned our model specifically for text-to-3D generation using the pre-trained weights, removing redundant prompts and freezing non-mesh textual embeddings. With a learning rate of 1e-5, context size of 1536, batch size of 4, gradient accumulation of 2, and 5 epochs of training, the fine-tuned model lost its general text capabilities but ,as shown in the Table 7, achieved text-to-3D results comparable to Trellis—demonstrating the inherent difficulty of balancing multiple tasks within a unified framework.

2) Trellis is built on a Rectified Flow (diffusion) architecture, while our model adopts a discrete autoregressive design. Diffusion and flow-based models currently hold an inherent advantage in visual generation quality, and surpassing them with autoregressive architectures remains an open research challenge. Nonetheless, our focus lies not in pushing the absolute performance of autoregressive models, but in enabling unified 3D generation and understanding under this paradigm—an innovative and promising direction as autoregressive visual generation continues to advance.

## 5   Conclusion

In this work, we introduce ShapeLLM-Omni, a novel framework that advances both 3D generation and understanding through a 3D VQVAE. By constructing a comprehensive 3D-Alpaca dataset, we provide a data foundation to support future research on native 3D-modality large language models.

**Limitation**   Constrained by limited resources, we possess only 70k 3D-editing pairs—far too few to achieve ChatGPT-4o–level results in 3D editing. Due to limited computing resources, our ShapeLLM-Omni only has 7B parameters. As a result, our performance hasn't yet reached the level of a true "3D version of ChatGPT-4o".

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

# A    More Experiments

## A.1    More Implementation details

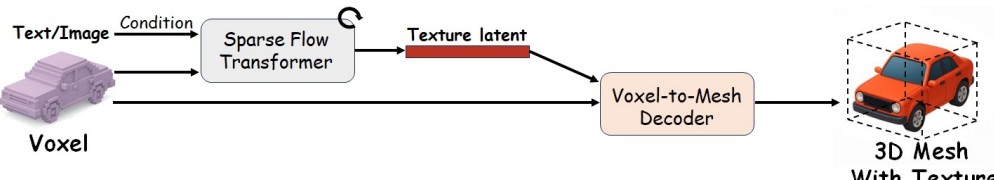

Figure 7: **About how to generate 3d mesh from voxel.** This image illustrates the process of reconstructing a textured mesh from voxel inputs using a texture transformer Xiang et al. [2024] and mesh decoder.

**Decoding Voxel into 3D Mesh**    As illustrated in the upper part of Figure 7, we first utilize a texture transformer, named Sparse-Flow Transformer Xiang et al. [2024], to extract texture latents from the voxel representation. These latents are then fed into a voxel-to-mesh decoder, which generates a mesh with associated texture information. Interestingly, we observe that the geometry of the output mesh is entirely determined by the input voxel representation, regardless of the presence of texture information.

**More Details about Training**    The model is trained on 48 H100 GPUs for 60k iterations. We conduct full parameter fine-tuning. We use the AdamW optimizer, with a learning rate of 1e-5, a warm-up of 400 steps with cosine scheduling, and a global batch size of 192. The total training time is around 5 days.

## A.2    More details about 3D-Alpaca

**3D Editing Prompt List**    As shown in Table 13 and Table 14, we present 70 out of the 100 categories from the 3D editing dataset, along with their corresponding editing prompts.

**3D Editing Data**    As shown in Figure 10, we present several examples from our 3D editing dataset. The figure illustrates that our 3D editing data pairs support effective modifications while preserving subject consistency between the original and edited versions.

## A.3    More Qualitative comparisons

In Figure 11, Figure 12, and Figure 13, we showcase additional Image-to-3D generation results. To maintain consistency with the training setup, all input images are resized to 512×512 resolution with a white background. This preprocessing step is crucial, as our base model, Qwen-VL Bai et al. [2025], encodes images into token sequences whose length depends on the input resolution. Additional Text-to-3D generation examples are presented in Figure 14. The visual results clearly demonstrate that our model is capable of producing high-fidelity 3D assets through a unified architecture. Furthermore, Figure 9 provides additional 3D-to-caption generation results, and Figure 8 shows two caption examples from Objaverse Deitke et al. [2023a]. The generated captions demonstrate that our ShapeLLM-Omni exhibits robust 3D understanding capabilities.

## A.4    More Quantitative comparisons

**Image-to-3D**    To provide a more comprehensive comparison, we quantitatively evaluate the image to 3D generation performance of OpenLRM, LGM, InstantMesh, and Unique3D on the same test set used in our paper. We also adopt the same evaluation metrics, including CLIPScore, $FD_{incept}$, and $KD_{incept}$. As shown in the table 8, our method consistently outperforms all baselines and demonstrates superior 3D generation quality.

**Evaluation of GPT-Annotated Dataset**    To quantitatively assess the quality of our GPT-annotated text-to-3D dataset, we randomly sample 1000 text–image pairs and evaluate their semantic alignment

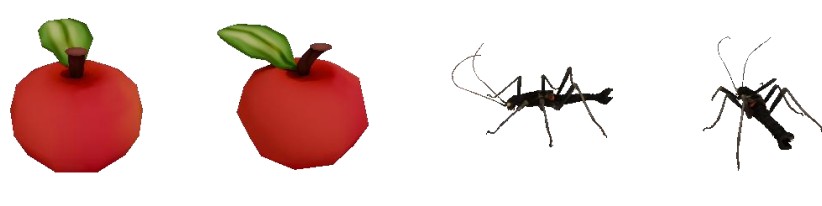

| UID | 0ea33b6617174530b97d6b7a92c275fb | de8ec2a724f14fc4b54624512f80f13e |
|---|---|---|
| InstructBLIP | An appleavatar 3d model | A black insect |
| 3D-LLM | A 3D model of a red apple. | A small, black spider with a long tail. |
| PointLLM | This is a 3D model of a unique apple, distinctively adorned with a single, vibrant green leaf at the top. | This 3D model depicts a realistic, jet-black insect with a pair of striking, golden brown eyes. |
| **ShapeLLM-Omni** | An apple with a stem and leaf. | A spider with multiple legs and a segmented body |

Figure 8: **Qualitative results on Objaverse.**

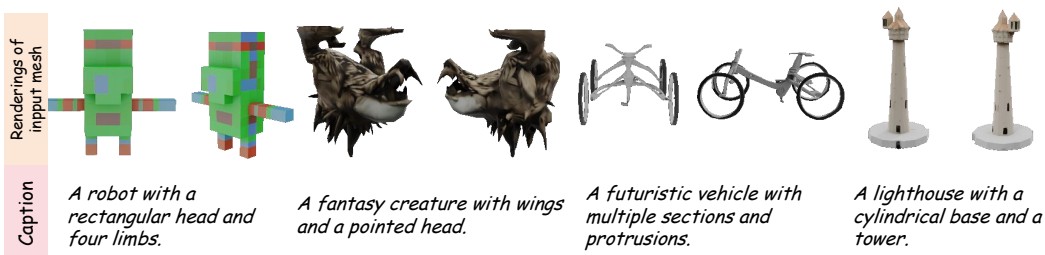

Renderings of input mesh

Caption

A robot with a rectangular head and four limbs.

A fantasy creature with wings and a pointed head.

A futuristic vehicle with multiple sections and protrusions.

A lighthouse with a cylindrical base and a tower.

Figure 9: **Some cases of 3D-to-caption result from our method.**

using CLIPScore and ViLT R-Precision. As shown in the Table 9, our dataset achieves high scores across both metrics, indicating strong correspondence between the captions and 3D content.

**Evaluation of 3D Editing Dataset**  To evaluate the alignment in 3D edting dataset, we compute the same metrics between the editing prompts and the rendered front-view of edited objects. The results, summarized in Table 10, indicate the editing prompts are well aligned with the resulting modifications.

**Evaluation of QA**  We conduct additional experiments to evaluate our model on 3D question-answering tasks. As shown in the table below, the best scores for each metric in the Table 11 are highlighted. Our model achieves consistently superior performance across all methods, demonstrating its strong capabilities in 3D QA tasks.

## A.5   Ablation

**3D VQVAE**  To determine the optimal codebook size for our 3D VQVAE model, we train several variants with different codebook sizes. We randomly sample 1000 meshes from the test dataset, voxelize them, and encode them into discrete tokens using each model. These tokens are then decoded into voxel grids and converted back to meshes through a voxel-to-mesh decoder. We evaluate reconstruction quality using Chamfer Distance (CD) and Hausdorff Distance (HD). As shown in Table 12, larger codebooks lead to better reconstruction performance. However, the improvement levels off beyond a codebook size of 8192, indicating saturation. We therefore choose 8192 as the final codebook size to strike a balance between quality and efficiency.

Table 8: More quantitative comparison of image-to-3D generation performance across OpenLRM, LGM, InstantMesh, and Unique3D on the shared test set, evaluated using CLIPScore, $FD_{incept}$, and $KD_{incept}$.

| Model | CLIP↑ | FD$_{incep}$ ↓ | KD$_{incep}$ ↓ |
|---|---|---|---|
| Instantmesh Xu et al. [2024a] | 74.50 | 22.0 | 0.25 |
| OpenLRM He and Wang [2023] | 72.75 | 20.1 | 0.26 |
| Unique3D Wu et al. [2024a] | 77.10 | 16.8 | 0.13 |
| LGM Tang et al. [2024a] | 75.20 | 18.5 | 0.15 |
| ShapeLLM-Omni(Ours) | **84.50** | **12.2** | **0.09** |

Table 9: Quantitative evaluation of the GPT-annotated text-to-3D dataset using CLIPScore, ViLT R-Precision@5 and ViLT R-Precision@10.

| | CLIPScore ViLT | R-Precision R@5 | R-Precision R@10 |
|---|---|---|---|
| Our | 29.58 | 35.3 | 42.2 |

Table 10: Quantitative evaluation of the GPT-annotated 3D Editing dataset using CLIPScore, ViLT R-Precision@5 and ViLT R-Precision@10.

| | CLIPScore ViLT | R-Precision R@5 | R-Precision R@10 |
|---|---|---|---|
| Our | 27.41 | 33.6 | 43.5 |

Table 11: Performance comparison on 3D question-answering tasks.

| Model | B-1 | R-L | METEOR | S-BERT | S-CSE |
|---|---|---|---|---|---|
| ShapeLLM Qi et al. [2024a] | 17.73 | 19.91 | 21.86 | 51.32 | 50.95 |
| GPT4Point Qi et al. [2024b] | 6.51 | 8.59 | 5.80 | 29.66 | 32.18 |
| PointLLM-13B Xu et al. [2024b] | 17.23 | 19.70 | 20.48 | **52.66** | 53.21 |
| **ShapeLLM-Omni (Ours)** | **19.66** | **21.31** | **22.68** | 52.51 | **53.33** |

Table 12: **Ablation study on the codebook Size of 3D VQVAE**

| Vocabulary Size | Chamfer Distance↓ | Hausdorff Distance↓ |
|---|---|---|
| 4096 | 0.0102 | 0.0561 |
| 8192 | **0.0094** | **0.0525** |
| 16384 | 0.0095 | 0.0534 |

Table 13: Edited Prompt Collection: Part One

| ID | Category | Edited prompt |
|---|---|---|
| 1 | Car | Add a cannon to the front, Open the door, Add a roof rack, Add a rear wing, Lengthen the car body, Shorten the car body, Convert into a convertible, Change wheels to square shape, Bend the roof, Add air vents on the sides, Install a spotlight on the roof, Open the hood, Install a rear-view camera |
| 2 | Tricycle | Add a wheel, Install a small trumpet |
| 3 | Bicycle | Raise the seat, Add a wheel, Install a basket |
| 4 | Traffic light | Add an extra light, Install a surveillance camera |
| 5 | Spaceship | Add wings, Add jet flames, Add solar panels, Install radar antenna, Shorten fuselage, Bend the tail fins downward, Bend the tail fins upward, Widen the wingspan, Narrow the wingspan, Tilt the whole body, Mount small missiles on wings |
| 6 | Tank | Rotate cannon to the side, Mount a telescope on the turret top |
| 7 | Character | Raise both hands, Raise left hand, Raise right hand, Hold a sword, Enlarge the head, Sit cross-legged, Wear a backpack, Wear a shoulder bag, Change to running pose, Grow a pair of wings, Stand on wind-fire wheels, Step on rocket launchers, Wear glasses, Wear a tall hat, Spread arms, High knee movement, Stand on one leg, Add a cape, Hold a shield, Grow a tail, Twist the waist, Stand on a skateboard, Change hairstyle to a bun, Enlarge the ears, Bend the elbows, Wear armor, Kneel on both legs, Cross both arms, Add halo above the head |
| 8 | Robot | Turn feet into wheels, Turn hands into bayonets, Wear an Iron Man helmet, Lengthen the arms, Mount mechanical wings on the back, Add antenna to the head, Add springs to the soles, Mount a rocket booster on the back, Lengthen the legs, Turn hands into cannons, Turn hands into claws, Turn arms into chainsaws, Add solar panels to the back, Transform into spider legs |
| 9 | Table | Put a vase on the table, Change table shape to round, Lay a tablecloth, Spiral-shaped table legs, Add a drawer under the tabletop, Jagged edges on the tabletop, Dig a hole in the center, Put a cup on the table, Add wheels under table legs, Put a fruit plate on the table |
| 10 | Chair | Place a cushion, Extend the legs, Shorten the legs, Add wheels to the feet, Install a footrest, Place a seat cushion, Add storage bags on the sides, Put a speaker on it, Turn into a rocking chair |
| 11 | Cabinet | Add cabinet doors, Open the cabinet doors, Add drawers, Pull out a drawer, Put a table lamp on top, Add a lock, Add internal shelves, Place a potted plant on top Bowl: Change to square, Put an egg inside, Add a pair of chopsticks |
| 12 | Bed | Add a pillow, Change to round shape, Add bed curtains, Place a kitten on the bed, Convert into a bunk bed |
| 13 | Sofa | Place a blanket, Place a teddy bear, Add a throw pillow |
| 14 | Bowl | Change to square, Put an egg inside, Add a pair of chopsticks |
| 15 | Backpack | Transform into a jetpack, Transform into a rolling backpack |
| 16 | Gun | Lengthen the barrel, Add barrels on both sides, Mount a scope on top, Add a magazine slot on the left, Attach a bayonet under the muzzle |
| 17 | Shoes | Extend the upper part, Thicken the sole, Attach wind-fire wheels |
| 18 | Clothes | Convert to short-sleeve, Convert to long-sleeve, Add a scarf |
| 19 | Hat | Raise the crown, Add wings to the sides, Turn the top into animal ears |
| 20 | Glasses | Change to round frames, Add a head strap, Remove the frames |
| 21 | Ring | Add a diamond, Remove the diamond |
| 22 | Knife | Extend the blade, Turn into "Zangetsu" from Bleach |
| 23 | Sword | Lengthen the blade, Wrap the blade in flames, Make the blade serrated, Add a ring guard to the hilt, Embed gems in the blade |
| 24 | Teapot | Change the spout length, Open the lid, Turn the spout into a chainsaw, Add a heater at the bottom |
| 25 | Bottle | Only upper half remains, Insert a rose, Pour tea into the bottle, Replace cap with cork, Tie a label around the neck |
| 25 | Cup | Turn into conical flask, Add a handle, Add a lid, Insert a straw, Add a cup heater |
| 26 | Cat | Jumping pose, Skating on a skateboard, Add a pair of wings, Wear clothes, Wear a bow on the head |
| 27 | Dog | Hold a bone in mouth, Add a dog leash, Wear clothes, Wear a Christmas hat |
| 28 | Insect | Remove wings, Remove antennae, Add an antenna, Add a pair of wings |
| 29 | Fish | Wear goggles |
| 30 | Block-shaped Object | Be stretched |
| 31 | Ball-shaped Object | Change to oval |

Table 14: Edited Prompt Collection: Part Two

| ID | Category | Edited prompt |
|---|---|---|
| 32 | House | Add chimney on roof, Add and open a door, Change roof to dome, Change door to arch, Add canopy on the door, Add garage on the side, Add a balcony, Add a street lamp next to house, Add a fence, Add a mailbox at entrance, Install solar panels on roof |
| 33 | Tower | Shorten height, Add flag on top, Add door at base, Add spotlight at tip, Add fence around, Add antenna on top, Add spiral staircase outside, Add window in middle, Add vines on surface, Keep only lower half, Add observation deck at top |
| 34 | Tree | Grow two giant hands, Grow giant flowers on top, Grow stars at top, Grow two long legs, Grow large wings on sides, Butterfly perching on tree, Add a door on trunk, Hang lanterns on branches |
| 35 | Flower | Add more petals, Insert into vase, Bee perching on it |
| 36 | Fruit | Put in fruit plate, Peel skin, Insert small umbrella on surface |
| 37 | Vegetable | Be stretched |
| 38 | Phone | Turn into tri-fold screen, Add stylus on edge |
| 39 | Computer | Grow wheels |
| 40 | TV | Add two antennas, Install base stand |
| 41 | Keyboard | Change to round keycaps |
| 42 | Book | Grow two arms and legs, Grow wings |
| 43 | Building | Add arched entrance in front, Install antenna on roof, Add chimney on roof, Add external staircase, Add billboard on top, Helicopter parked on roof, Add fence in front, Make building round, Install solar panels on roof, Add flag on roof, Change door to revolving door, Add a clock on wall, Hang string lights on wall |
| 44 | Building Structure | Remove one column, Change to flat roof, Convert to castle top, Add cable support structure |
| 45 | Statue | Add a pair of wings, Wear sunglasses, Wear headphones, Wear a tall hat, Add halo above, Add fence around, Add multiple arms, Change head to Medusa, Wear a flower crown, Be wrapped in chains |
| 46 | Lamp | Change bulb to square, Change lampshade shape, Add more lamp heads, Change lamp head direction, Add hanging chains |
| 47 | Door | Replace rectangle door with arch, Add doorbell, Add surveillance camera, Add door lock, Add steps at entrance, Open the door, Wrap door with vines, Add peephole Bird: Claw grasping branch, Wings spread, Pecking downward, Lengthen beak, Shorten beak, Wear top hat, Hold a branch in beak, Wear goggles |
| 48 | Sculpture | Wear crown, Wear armor, Wear mask, Hold scepter |
| 49 | Weapon | Add hook at front, Make blade wavy, Change to double-headed, Be chained |
| 50 | Helmet | Add goggles, Add visor, Change to pointed top, Unfold side wings |
| 51 | Bridge | Convert to suspension bridge, Add pillars, Make multi-level, Add street lights, Add toy cars |
| 52 | Vase | Insert flowers, Place on table, Add handles on sides |
| 53 | Mechanical Arm | Replace hand with clamp, Arm rotates |
| 54 | Plant | Add fruits, Broken branches, Grow upwards |
| 55 | Shield | Change to octagonal, Embed gem in center, Insert an arrow, Wrap in vines |
| 56 | Chest | Be flattened, Open lid, Lock with chains |
| 57 | Airplane | Mount missiles under wings, Retract landing gear, Extend landing gear, Add more engines |
| 58 | Castle | Add drawbridge at entrance, Attach a dragon on wall, Connect towers with bridges, Hang flags on walls |
| 59 | Mythical Creature | Add saddle, Grow spikes on back, Sleep curled on ground |
| 60 | Pillar | Change to polygonal, Bend to one side, Add grooves to body |
| 61 | Tool | Lengthen handle, Replace tool head with bayonet, Bend the handle |
| 62 | Lighthouse | Add radar antenna on top, Add spiral staircase outside, Add window |
| 63 | Box | Be flattened, Open the lid, Punch a hole |
| 64 | Monument | Change top to pointed, Add flag on top, Add steps at base |
| 65 | Animal | Grow antennae |
| 66 | Stairs | Add more steps, Change to spiral stairs, Remove handrails |
| 67 | Tent | Extend awning, Change to dome-shaped |
| 68 | Street Light | Add signboard on pole, Add camera on pole |
| 69 | Trophy | Add lid, Add handles |
| 70 | Machine | Add wheels |

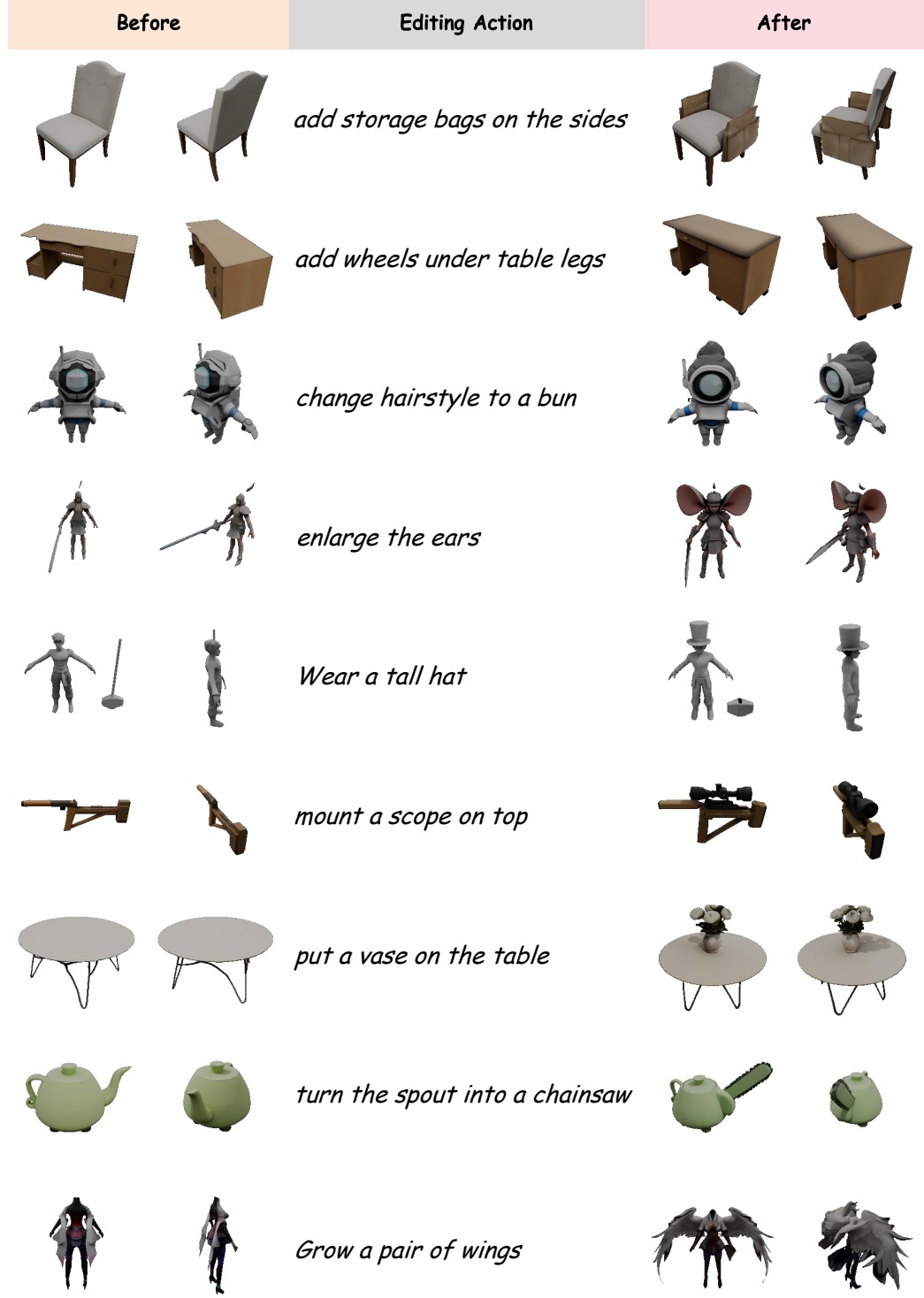

Figure 10: **Some cases of our 3D-Editing Data**

| Input | 3D Mesh Output |
|---|---|

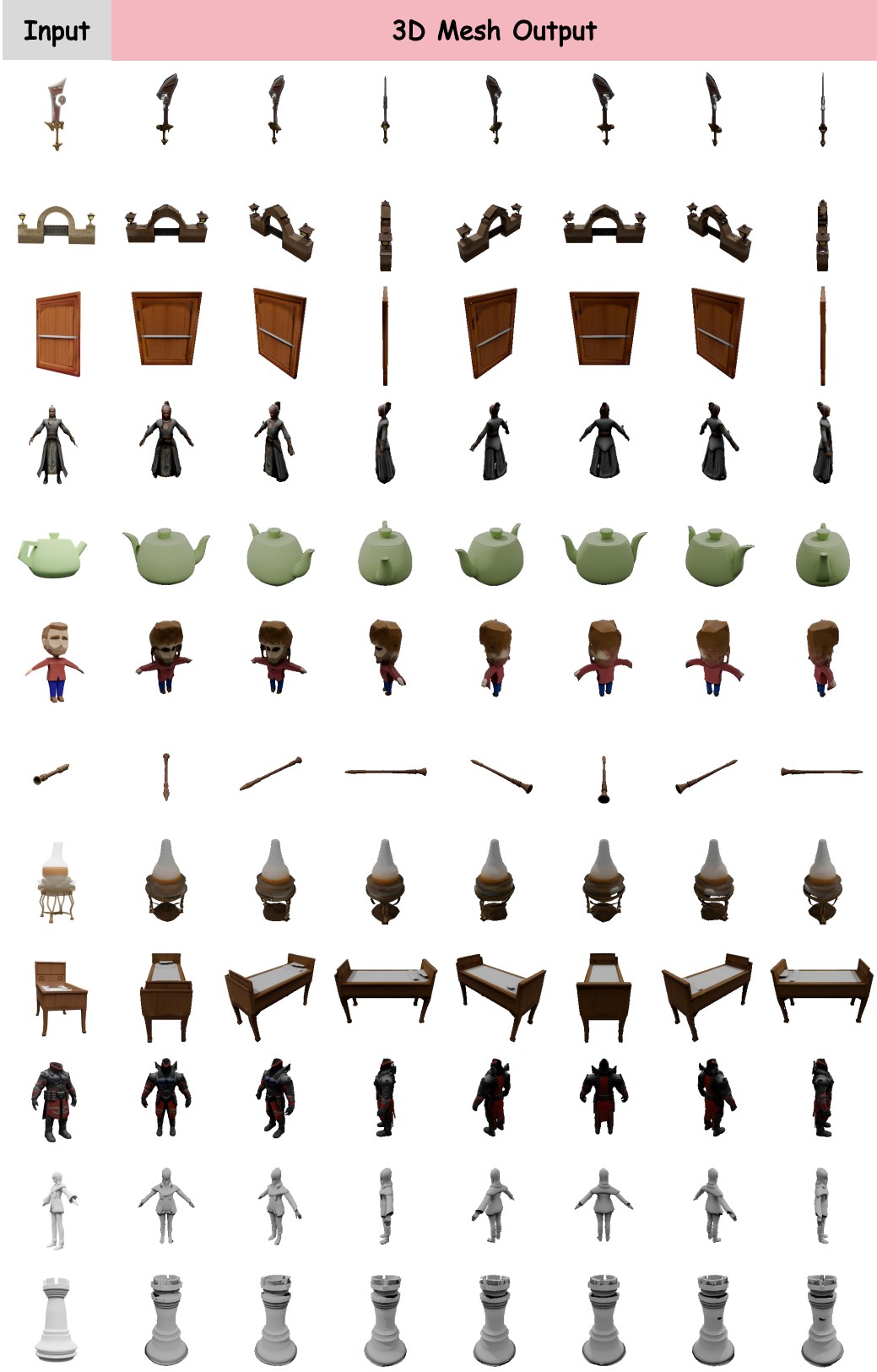

Figure 11: **More cases of Image-to-3D result from our method.**

| Input | 3D Mesh Output |
|---|---|

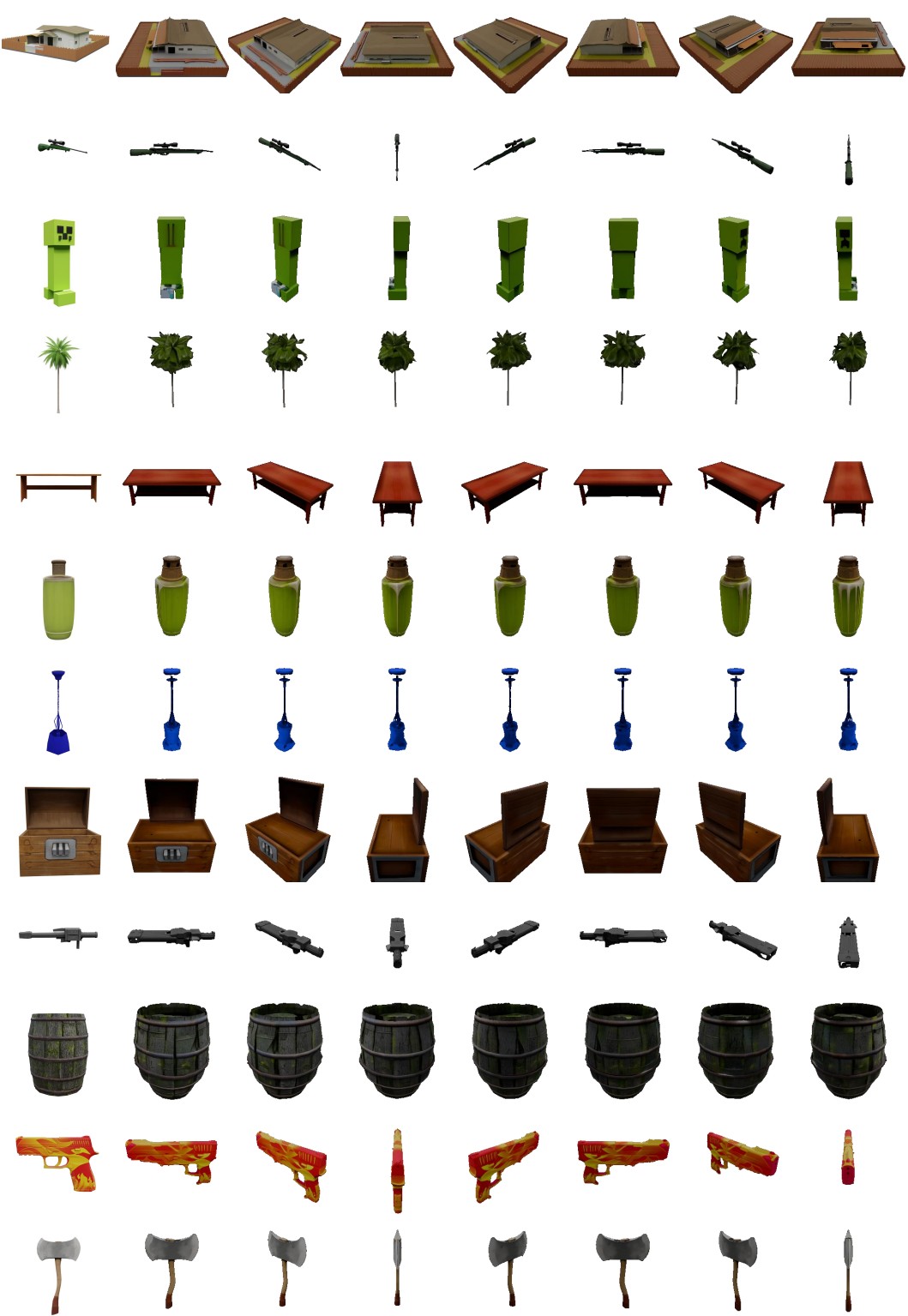

Figure 12: **More cases of Image-to-3D result from our method.**

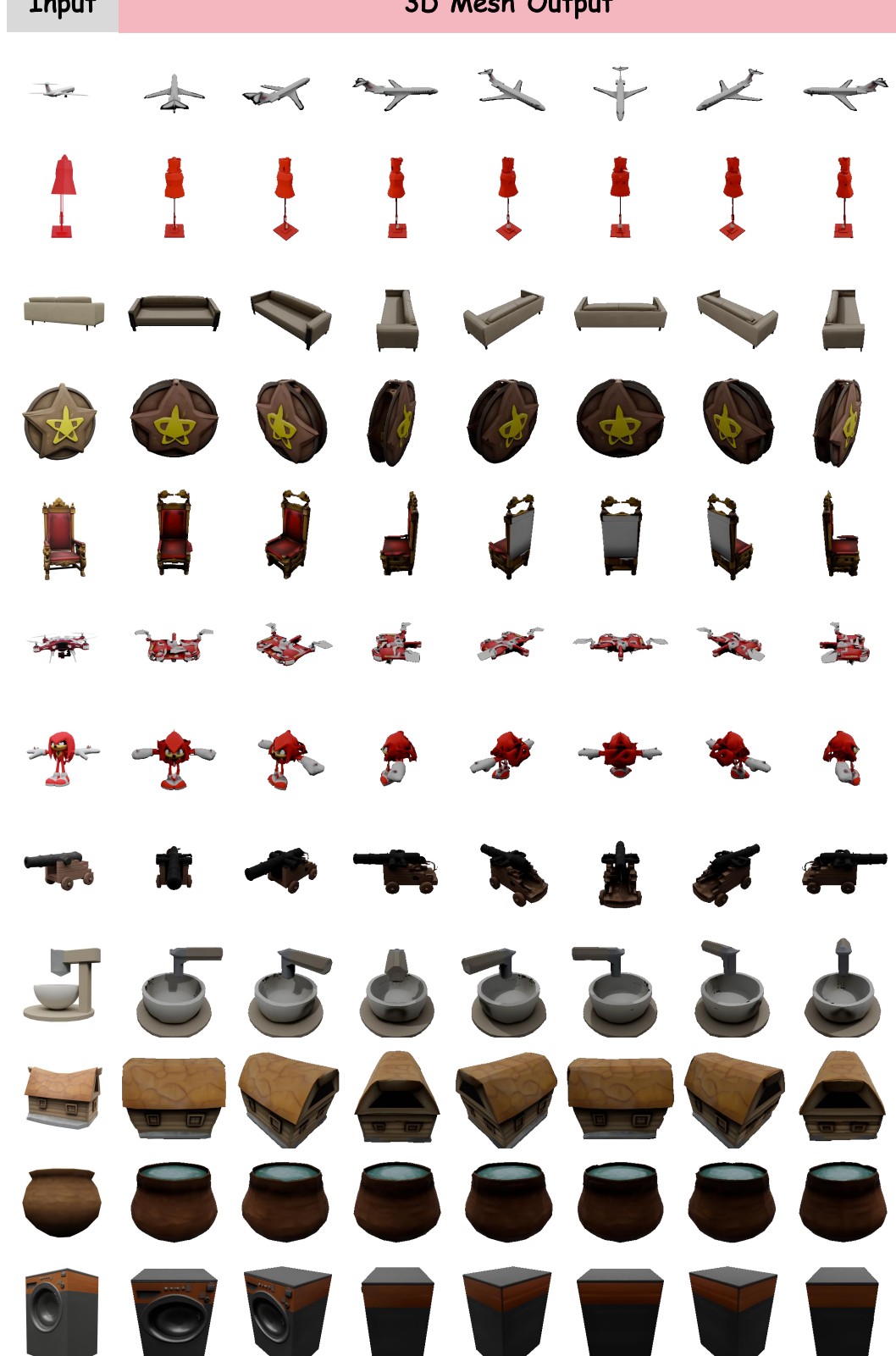

Figure 13: **More cases of Image-to-3D result from our method.**

| Input | 3D Mesh Output |
|-------|----------------|

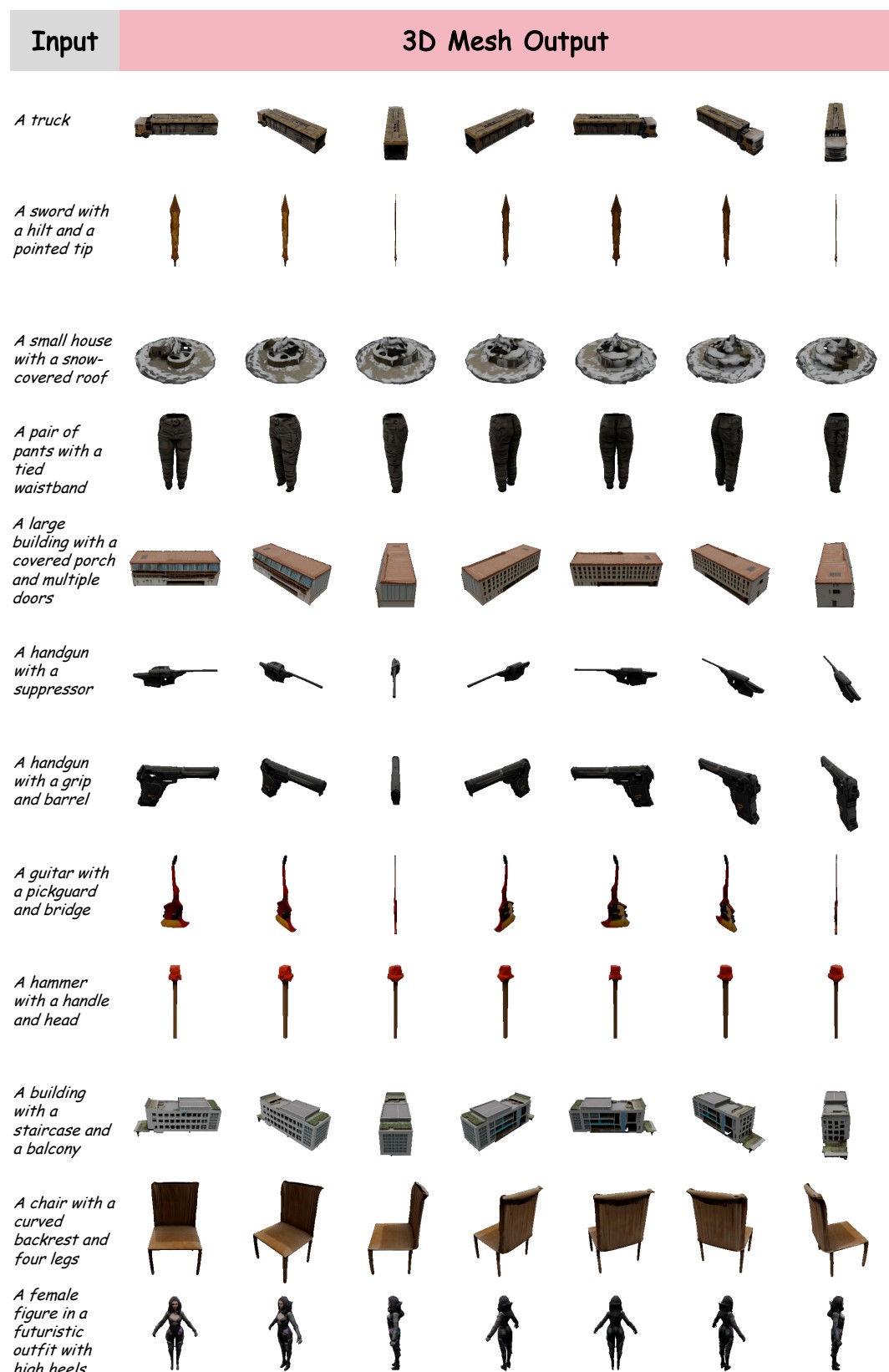

Figure 14: **More cases of Text-to-3D result from our method.**

