# OpenReview forum: "ShapeLLM-Omni: A Native Multimodal LLM for 3D Generation and Understanding"
_NeurIPS.cc/2025/Conference — NeurIPS 2025 spotlight_

### Official Review · Reviewer_Vmok · 2025-07-01

**Clarity:** 3
**Significance:** 3
**Originality:** 3
**Rating:** 4
**Confidence:** 4

**Summary:**

This paper proposes ShapeLLM-4o, a unified 3D-language model. As the first unified multi-task 3D understanding and generation framework, ShapeLLM-4o incorporates multiple 3D-related tasks like text-to-3D, image-to-3D and 3D editing. By introducing a 3D-VQVAE, 3D assets are aligned to the vision and language tokens in a LVLM. To train the unified model, the paper raises a comprehensive 3D-language dataset called 3D-Alpaca. The proposed ShapeLLM-4o achieves competitive results on a wide range of downstream tasks.

**Questions:**

1. The paper attributes ShapeLLM-4o’s underperformance on 3D generation tasks (compared to Trellis) to its multi-task design, while Trellis uses task-specific models. However, have the authors experimented with single-task fine-tuning of ShapeLLM-4o?


2. Why does the paper omit evaluation on 3D visual question answering (VQA), which is a more comprehensive test of the model’s language understanding than captioning? There are previous works that evaluate 3D LVLMs on QA tasks[a], [b], but we did not see it in ShapeLLM-4o.

[a] ShapeLLM: Universal 3D Object Understanding for Embodied Interaction, ECCV24

[b] GPT4Point: A Unified Framework for Point-Language Understanding and Generation, CVPR24

**Ethical Concerns:**

["NO or VERY MINOR ethics concerns only"]

**Final Justification:**

My question has been well resolved.

**Limitations:**

The authors have addressed the limitations.

**Paper Formatting Concerns:**

There is no formatting issue in this paper,

**Quality:**

3

**Strengths And Weaknesses:**

Strengths

1.	ShapeLLM-4o serves as the first unified model that can tackle text-to-3D, image-to-3D, 3D-to-3D (editing) and 3D-to-text tasks in all. By integrating a wide range of practical tasks, it makes a worth noting contribution to 3D understanding and generation.

2.	The proposed 3D-Alpaca dataset is a good advancement to the community, in both its data scale and task comprehensiveness. The curation of 3D data is challenging, but the paper smartly utilizes existing expert models to create high-quality data.

3.	The paper writing is clear and organized, making the method easy to follow.

Weaknesses

1.	Although supporting multiple tasks, ShapeLLM-4o underperforms Trellis on 3D generation tasks. It is stated in the paper that a possible reason is Trellis trains separate models for different tasks. However, what is the result of ShapeLLM-4o if further fine-tuned on a single task?

2.	ShapeLLM-4o is evaluated on 3D caption tasks, but why not include 3D question-answering? Captioning could not fully reveal the language power of the model. There are previous works that evaluate 3D LVLMs on QA tasks[a], [b], but we did not see it in ShapeLLM-4o.

[a] ShapeLLM: Universal 3D Object Understanding for Embodied Interaction, ECCV24

[b] GPT4Point: A Unified Framework for Point-Language Understanding and Generation, CVPR24

3.	What is the cost of building the 3D editing data in 3D-Alpaca?

4.	What is the number of 3D editing data? In Tab.2 it is 420k, but in the ‘Limitations’ part it is stated as 70k.

---

> ### Author Rebuttal · Authors · 2025-07-31
>
> We thank the reviewers for the constructive reviews. We provide our feedbacks as follows.
>
> > **Weaknesses1**
>
> **A1:** Based on the pretrained weights, we conducted a small-scale fine-tuning experiment specifically targeting text-to-3D generation. During fine-tuning, we removed redundant prompt tokens. Additionally, we froze the gradients of all text token embeddings except for the mesh tokens. The learning rate was set to 1e-5, with a context length of 1536, batch size of 4, and a gradient accumulation step of 2. The fine-tuning lasted for 5 epochs.
> After fine-tuning, the model lost general text processing ability but improved significantly on text-to-3D tasks, achieving results comparable to Trellis on our test dataset. This shows the model can gain on a specific task by sacrificing general capabilities.
> We think Shapellm-4o did not outperform Trellis because diffusion/flow models currently work better than autoregressive models for visual generation. Beating diffusion/flow with autoregressive models is a major challenge. Our work focuses not on maximizing 3D generation performance but on creating a unified model that can both understand and generate 3D content autoregressively. This is a key innovation. As autoregressive visual generation advances, our unified model’s performance will also improve.
>
> | **Method**   | **CLIP↑** | **FDincep↓** | **KDincep↓** |
> |:------------:|:-----------:|:------------:|:-----------:|
> | Trellis     |   30.8   |   18.3   |   0.19 |
> | ShapeLLM-4o|   26.7   |   25.9   |   0.25 |
> | ShapeLLM-4o（Overfitting）|   30.1   |   18.9   |   0.21 |
>
> > **Weaknesses2**
>
> **A2:** We conduct additional experiments to evaluate our model on 3D question-answering tasks. As shown in the table below, the best scores for each metric in the table are highlighted. Our model achieves consistently superior performance across all methods, demonstrating its strong capabilities in 3D QA tasks.
>
> | **Method**   | **BLUE-1↑** | **ROUGE-L↑** | **METEOR↑** | **Sentence-BERT↑** | **SimCSE↑** |
> |:------------:|:-----------:|:------------:|:-----------:|:-----------------:|:-----------:|
> | ShapeLLM     |   17.73   |   19.91   |  21.86 |       51.32      |     50.95     |
> | GPT4Point    |     6.51     |     8.59      |     5.80     |       29.66      |     32.18     |
> | PointLLM-13B |     17.23    |     19.70     |     20.48    |     **52.66**    |   53.21  |
> | ShapeLLM-4o  |  **19.66**  |   **21.31**   |  **22.68**   |   52.51   |   **53.33**   |
>
> > **Weaknesses3**
>
> **A3:** We used the GPT-Image-1 API for generating and editing the data, selecting the "low" quality option. The average cost per image editing pair was around 0.01, with a total expenditure of approximately 750. After constructing the image editing pairs, we used Trellis for 3D reconstruction of the edited images. On average, the time to generate one asset was 5 seconds, and the total GPU time consumed was about 100 hours, which equals approximately 200. Therefore, the total cost for the entire process amounted to around 950.
>
> > **Weaknesses4**
>
> **A4:** We apologize for the confusion. As described in Section 3.4 of the paper, we initially constructed 70K 3D editing text-image pairs. Based on these, we generated a total of 420K multi-turn dialogue samples following specific rules. Specifically, for each 3D editing pair, we created three dialogue types: (1) pure 3D editing, (2) image-to-3D followed by 3D editing, and (3) text-to-3D followed by 3D editing—yielding approximately 210K dialogues. To further enrich the diversity and support multi-task interleaving, we inserted additional pure text instructions into these dialogues, ultimately expanding the dataset to 420K dialogue samples. We will revise the text in the ‘Limitations’ section to clearly reflect that 70K is the number of base 3D editing pairs, while 420K refers to the full dialogue dataset derived from them.

---

> > ### Comment · Reviewer_Vmok · 2025-08-04
> > **Thanks for the reply**
> >
> > Thank you for your reply. My question has been well resolved.

---

> > > ### Author Response · Authors · 2025-08-04
> > > **thank**
> > >
> > > We are glad that your concerns have been addressed, and we would like to sincerely thank you for your thoughtful and constructive review.

---

### Official Review · Reviewer_zpad · 2025-07-01

**Clarity:** 3
**Significance:** 3
**Originality:** 3
**Rating:** 4
**Confidence:** 4

**Summary:**

This paper addresses the modality limitation of current unified multimodal large language models (MLLMs), such as GPT-4o, which primarily support only image and text. To overcome this, the authors propose ShapeLLM-4o, a native autoregressive MLLM with the capability to understand and generate 3D objects.

The model leverages a pretrained VQ-VAE for 3D object representations, using the encoder for object understanding and the decoder for generation. To enable 3D reasoning capabilities, the authors construct a large-scale dataset covering tasks such as understanding, generation, and editing of 3D content. The training is based on the Qwen-2.5-VL-7B-Instruct backbone, pushing the frontier of multimodal LLMs toward the 3D modality.

**Questions:**

- In constructing the 3D editing dataset, it is stated that the authors use image editing followed by 3D reconstruction. Does this mean the model is prompted with an input image and generates edited images from multiple viewpoints?
  - How is consistency across multiple views ensured during this image editing process?
  - What specific 3D reconstruction method is used to obtain the final 3D shape?
  - A detailed visualization of this pipeline (from image edit to 3D generation) would be very helpful.
- The paper adopts Trellis’s 3D tokenizer while using a more powerful Transformer backbone (Qwen-2.5-VL), yet the 3D generation quality is significantly worse. However, the paper does not analyze this performance gap in detail.
- Is the main architectural difference between this work and Trellis simply replacing the flow transformer with a VLM backbone?

**Ethical Concerns:**

["NO or VERY MINOR ethics concerns only"]

**Final Justification:**

- The paper presents the first autoregressive framework that unifies 3D object understanding and generation, demonstrating the feasibility of extending LLMs to the 3D domain.
- It contributes a large-scale dataset tailored to training unified 3D-capable MLLMs.
- The study offers an exploratory step toward expanding the modality range of general-purpose MLLMs.

**Limitations:**

yes

**Paper Formatting Concerns:**

- Line 12–13 contains an incomplete sentence and should be revised for clarity.
- Line 153 has a typo: “ppaired” → “paired”.

**Quality:**

3

**Strengths And Weaknesses:**

**Strengths**
- The paper presents the first autoregressive framework that unifies 3D object understanding and generation, demonstrating the feasibility of extending LLMs to the 3D domain.
- It contributes a large-scale dataset tailored to training unified 3D-capable MLLMs.
- The study offers an exploratory step toward expanding the modality range of general-purpose MLLMs.

**Weaknesses**
- The idea of using a shared tokenizer for understanding and generation has already been validated extensively in the vision domain (e.g., EMU3), and is not novel in principle.
- In 3D generation visualizations, the results show noticeable inconsistencies compared to baselines like Trellis (e.g., in the third row, the number of buttons on the TV is incorrect).
- In the 3D understanding evaluation, comparisons are made only with PointLLM, which is relatively outdated. More recent baselines should be included for a fair assessment.

---

> ### Author Rebuttal · Authors · 2025-07-31
>
> We thank the reviewers for the constructive reviews. We provide our feedbacks as follows.
>
> > **Weaknesses1**
>
> **A1:** While prior works such as Emu3 have demonstrated the effectiveness of using a shared tokenizer for both understanding and generation in the 2D vision domain, extending this paradigm to 3D data is far from straightforward. In general, mapping modality-specific features into a unified language space is a powerful and broadly applicable idea—one that spans across modalities from text to vision. However, each modality presents its own challenges. In the case of 3D, the representation is structurally richer and requires maintaining spatial integrity and multi-view consistency, which makes joint modeling significantly more difficult than in 2D. Our contribution lies in successfully adapting this framework to the 3D object domain by employing a new voxel-based VQ-VAE, and empirically validating its effectiveness on both generation and understanding tasks. We appreciate the reviewer’s insight and will incorporate this discussion into the related work section to better contextualize the novelty of our approach.
>
> > **Weaknesses2**
>
> **A2:** We sincerely thank the reviewer for the careful observation. The inconsistency arises primarily form differencies in data preprocessing: our original setting does not normalize the object occupancy within the input image, whereas Trellis resizes the object to occupy a fixed portion. This discrepancy results in an unfair comparison. After aligning with Trellis, our results demonstrate significantly improved visual consistency in the mentioned failure examples. We will include this updated setting and analysis in the final version to ensure a fair and rigorous comparison.
>
> > **Weaknesses3**
>
> **A3:** We conduct additional 3D understanding comparisons with recent and stronger methods, including ShapeLLM and GPT4Point. As shown in the table below, the best and second-best results for each metric are marked in bold and underline, respectively. Our method consistently outperforms these baselines, achieving state-of-the-art performance in 3D understanding tasks.
>
> | **method**       | **BLUE-1↑** | **ROUGE-L↑** | **METEOR↑** | **Sentence-BERT↑** | **SimCSE↑**   |
> |:--------------:|:-----------:|:------------:|:-----------:|:------------------:|:-------------:|
> | ShapeLLM       | 17.88| 19.24        | 17.96| 48.52              | 49.98         |
> | GPT4Point      | 8.45        | 10.11        | 13.13       | 40.31              | 42.88         |
> | PointLLM-13B   | 17.09       | 20.99 | 16.45       | **50.15**          | **50.83**     |
> | ShapeLLM-4o    | **18.51**   | **21.37**    | **19.89**   | 49.34      | 50.72  |
>
> > **Questions1**
>
> **A4:** We thank the reviewer for this valuable suggestion. In our current pipeline, we first edit single-view image using GPT-4o, then perform image to 3D generation using Trellis. We agree that incorporating view-consistent editing and leveraging more large 3D reconstruction models could further improve the editing fidelity and geometric accuracy of the final 3D shapes. We will serve this suggestion as future direction in the conclusion section, and explore it in our future work.
>
> > **Questions2**
>
> **A5:** We have provided a preliminary discussion on the performance gap in Section 4.2, specifically in the third paragraph under the subsection "Compared with Trellis." We believe the performance gap between our model and Trellis can be attributed to two main reasons:
>
> **Reason 1:** Our model integrates six tasks (3D generation from images, 3D generation from text, 3D understanding, 3D editing, image understanding, and text reasoning) into a single model. In contrast, Trellis separates the tasks of 3D generation from images and 3D generation from text into two distinct models, each trained independently. The multi-task integration is inherently more challenging than training a single-task model. To demonstrate this, we conducted a small-scale fine-tuning experiment specifically for text-to-3D generation, using the existing pre-trained weights. During training, we removed extraneous prompt words, retaining only the basic prompt, and froze the gradients of the textual tokens corresponding to embeddings other than the mesh token. The hyperparameters used were: learning rate of 1e-5, context size of 1536, batch size of 4, and gradient accumulation of 2. We continued fine-tuning for 5 epochs. After fine-tuning, the model lost its ability to generate normal text outputs; however, its ability for text-to-3D generation improved significantly, yielding results on the test dataset comparable to those of Trellis. This experiment shows that our model, by sacrificing other capabilities, can achieve further breakthroughs in a single task.
>
> **Reason 2:** We believe a key factor preventing ShapeLLm-4o from surpassing Trellis is the inherent advantage of diffusion/flow models over autoregressive architectures in visual content generation. The diffusion/flow models currently outperform autoregressive models in visual generation tasks. Overcoming this gap, where autoregressive architectures can outperform diffusion/flow models, remains a significant challenge in the academic community.
> However, the focus of our work is not to explore the upper bounds of autoregressive architectures for 3D generation but to enable unified generation and understanding in a model. Our goal is for LLMs to successfully leverage autoregressive paradigms to both understand and generate 3D content. This is a significant innovation in the field. Moreover, autoregressive visual generation is a highly active area of research, and the performance of the unified model architecture we propose will only improve as advances in autoregressive visual generation continue.
>
> | **Method**   | **CLIP↑** | **FDincep↓** | **KDincep↓** |
> |:------------:|:-----------:|:------------:|:-----------:|
> | Trellis     |   30.8   |   18.3   |   0.19 |
> | ShapeLLM-4o|   26.7   |   25.9   |   0.25 |
> | ShapeLLM-4o（Overfitting）|   30.1   |   18.9   |   0.21 |
>
> > **Questions3**
>
> **A6:**  The main difference between our work and Trellis is not just swapping the flow transformer with a VLM backbone. There are two key distinctions:
> Generative Modeling Approach: Trellis uses a flow-based model, while our approach relies on an autoregressive model. These two methods are fundamentally different in how they generate data.
> Discrete vs. Continuous Representation: Trellis uses continuous latent spaces, whereas our model works with discrete tokens. This affects how the 3D content is represented and generated.
> So, the architectural difference is much more than just the backbone change — it's about the overall generative modeling approach and representation type.

---

> > ### Comment · Reviewer_zpad · 2025-08-04
> >
> > Thank you for the additional experiments and clarifications. I am willing to keep my score. In addition, please pay attention to the minor writing issues raised in order to improve the quality of the final manuscript.
> >
> > - Line 12–13 contains an incomplete sentence and should be revised for clarity.
> > - Line 153 has a typo: “ppaired” → “paired”.

---

> > > ### Author Response · Authors · 2025-08-04
> > > **thank**
> > >
> > > We are glad that your concerns have been addressed, and we would like to sincerely thank you once again for your thoughtful and constructive review. Thank you also for pointing out the typos. We will carefully revise and polish the paper upon acceptance.

---

### Official Review · Reviewer_LdZP · 2025-07-01

**Clarity:** 2
**Significance:** 2
**Originality:** 2
**Rating:** 4
**Confidence:** 2

**Summary:**

This paper introduces a new mllm called ShapeLLM-40. It mainly focuses on 3D generation and understanding. Compared with some existing models like GPT-4o, this model can handle 3D data together with text and image, so it can solve some limitations before. ShapeLLM-40 uses a 3D VQVAE to change 3D objects into discrete tokens. In this way, the model can process text, image and 3D data together in one framework. For training, the authors build a new dataset named 3D-Alpaca, which includes tasks like text-to-3D, image-to-3D, 3D caption and 3D editing. From the experiments, ShapeLLM-40 gets better results than baselines such as Trellis, SAR3D and 3DTopia-XL. It shows better text matching and generation quality, and also keeps good language and conversation ability.

**Questions:**

Please answer my concerns in the Weaknesses.

**Ethical Concerns:**

["NO or VERY MINOR ethics concerns only"]

**Final Justification:**

During the rebuttal, the authors have solved my concerns on why their method hasn't compare with scene-level method. However, my concern about their novelty on constructing language data hasn't been fully solved. Their proposed data engine highly overlapped with previous works like Grounded 3D-LLM, Chat-Scene, Robin3D, etc.

Given that, they are the preliminary work on doing both understanding and generation in 3D field. It is hard to confidently identify their contribution regarding novelty.

Above all, I raise my score to acceptance level but downweight my confidence score.

**Limitations:**

There's no obvious negative societal impact but for the limitations please refer to the Weaknesses.

**Paper Formatting Concerns:**

No.

**Quality:**

2

**Strengths And Weaknesses:**

Strengths:

1. Provide a good baseline to bridge both understanding and generation in 3D domain.
2. Contribution an valuable dataset to the community for future works.
3. The motivation to use 3D VQVAE for 3D modality encoding and decoding is clear and sound.

Weaknesses:

1. The 3D capability only focus on object-level, which limits its application in real life or contribution to 3D reasoning community. Currently, 3D scene level understanding is more realistic and challenging. Please consider to evaluate on 3D scene level benchmark for 3D captioning, QA, and Grounding. Related works include Robin3D, Chat-Scene, LL3DA, etc.
2. When applied to 3D scene-level reasoning, this 3D VQVAE would inevitable increase computational cost. As shown in Robin3D and Chat-Scene, a scene can be represented by 100 to 150 objects which means will scale up 100 to 150 times of the current number of vision tokens. I don’t think this is a practical approach.
3. The approach for generating 3D Generation and Understanding Dataset, which is actually generating paired-language data, is similar with Grounded 3D-LLM. Therefore, even though I value the engineering effort, there is indeed not much insight or novelty to the research field.
4. Most data are generated from GPT. There’s no guarantee on its quality since GPT will hallucinate. How to assess its quality would be a problem since data is the main contribution of this paper.
5. Given the huge training cost, especially 48 NVIDIA H100, and with a more advanced backbone Qwen-2.5-vl, the sub-optimal performance in Tab.4 shows that the proposed method is not very effective or efficient.

---

> ### Author Rebuttal · Authors · 2025-07-31
>
> We thank the reviewers for the constructive reviews. We provide our feedbacks as follows.
>
> > **Weaknesses1**
>
> **A1:** We emphasize that 3D object-level generation and understanding tasks are fundamentally different from scene-level ones in terms of objectives, input modalities, and methods. Due to this gap, models designed for object-level tasks often cannot be directly applied to scene-level benchmarks, and vice versa. This distinction is well acknowledged in the community—existing models such as PointLLM, GPT4Point, Trellis, and ShapeLLM are all tailored specifically for object-level tasks. This is not a limitation of the algorithms themselves, but a reflection of the fundamental differences between the two problem settings. Therefore, existing benchmarks designed for 3D scene understanding are not directly applicable to our method and do not offer a fair comparison.
> Moreover, object-level tasks serve as the foundation for 3D generation and understanding, since the ability to accurately generate and interpret 3D objects is critical for applications in robotics, virtual reality, gaming, and other domains. Robust object-level modeling is necessary for scaling to more complex 3D scene-level tasks.  We contend that ​the practical realism and methodological challenges inherent in object-level tasks are commensurate with those of 3D scene-level tasks.​
> Despite the importance, research combining unified autoregressive models with 3D object generation and understanding is few. Our work focuses on object-level tasks, marking an important step toward unifying 3D object generation and understanding within a generative modeling framework.
>
> > **Weaknesses2**
>
> **A2:** Our proposed unified autoregressive framework and VQ-VAE are specifically designed for object-level tasks, with tailored methodologies, data structures, and objectives not applicable to scene-level tasks. Therefore, directly employing existing scene-level 3D benchmarks to evaluate our framework is not only inappropriate but also fails to reflect the scope and advantages of our approach. Just as Trellis cannot be expected to generate complex scenes, our object-level trained model fundamentally lacks the architectural capacity for scene generation and understanding.
>
> > **Weaknesses3**
>
> **A3:** While our data construction is GPT-assisted, as in prior works, our main contribution lies not in proposing a novel data generation pipeline, but in constructing a critical but underexplored dataset. Specifically, Grounded 3D-LLM emphasizes scene-level understanding, whereas our dataset targets object-level editing. To the best of our knowledge, object-centric 3D editing data remains extremely scarce in existing work. We believe our dataset fills this important gap and offers a valuable resource for advancing research in controllable 3D generation and editing.
>
> > **Weaknesses4**
>
> **A4:**  To quantitatively assess the quality of our GPT-annotated text-to-3D dataset, we randomly sample 1000 text–image pairs and evaluate their semantic alignment using CLIPScore and ViLT R-Precision. As shown in the Table 1, our dataset achieves high scores across both metrics, indicating strong correspondence between the captions and 3D content.
>
> |            | **CLIPScore** | **ViLT R-Precision R@5** | **ViLT R-Precision R@10** |
> |:----------:|:-------------:|:------------------------:|:-------------------------:|
> | **Ours**   | 29.58     | 35.3                 | 42.2                 |
>
> Similarly, to evaluate the alignment in 3D edting dataset, we compute the same metrics between the editing prompts and the rendered front-view of edited objects. The results, summarized in Table 2, indicate the editing prompts are well aligned with the resulting modifications.
>
> |            | **CLIPScore** | **ViLT R-Precision R@5** | **ViLT R-Precision R@10** |
> |:----------:|:-------------:|:------------------------:|:-------------------------:|
> | Ours       | 27.41         | 33.6                     | 43.5                      |
>
> > **Weaknesses5**
>
> **A5:** Our framework jointly addresses multiple tasks—including 3D generation, understanding, and language capabilities—which inherently increases the training complexity and resource requirements. In particular, maintaining strong language capabilities while integrating diverse 3D tasks demands substantial computation. We verified it by conducting a fine-tuning experiment specifically targeting text-to-3D generation, based on the pretrained weights. During fine-tuning, we removed redundant prompt tokens. Additionally, we froze the gradients of all text token embeddings except for the mesh tokens. The learning rate was set to 1e-5, with a context length of 1536, batch size of 4, and a gradient accumulation step of 2. The fine-tuning lasted for 5 epochs. After fine-tuning, the model lost general text processing ability but improved significantly on text-to-3D tasks, achieving better results on our test dataset. This shows the model can gain on a specific task by sacrificing general capabilities.
>
> | **Method**   | **CLIP↑** | **FDincep↓** | **KDincep↓** |
> |:------------:|:-----------:|:------------:|:-----------:|
> | ShapeLLM-4o|   26.7   |   25.9   |   0.25 |
> | ShapeLLM-4o（Overfitting）|   **30.1**   |   **18.9**   |   **0.21** |

---

> > ### Comment · Reviewer_LdZP · 2025-08-02
> > **Clarification Request on Evaluation Scope and Methodological Novelty**
> >
> > Thank you for your detailed response.
> >
> > However, I would like to reiterate that your rebuttal and paper should be self-contained and convincing on their own, rather than relying on what other works like Trellis or GPT4Point have or have not done. Simply pointing out that existing object-level works also avoid scene-level evaluation does not justify why such a choice is appropriate for your work.
> >
> > If you claim that comparisons with scene-level benchmarks are unfair, I would suggest you to explain why in concrete, technical terms. From my understanding, the main distinction you draw between object-level and scene-level tasks appears to be in the number of points or the size of the input, rather than in the data modality or structure itself. In fact, prior works like Chat-Scene or Robin3D have demonstrated how object-level point clouds can be extracted from scene-level data using object detectors or masks. These can be formatted into the same input structure your model expects. Given this, I don't find it compelling to claim such evaluations are "inappropriate".
> >
> > I also appreciate your points about the importance of object-level modeling in downstream applications like robotics and VR. However, the paper is under review for its algorithmic contribution, not its application domain. Switching from 3D scene-level to object-level tasks does not itself constitute a technical innovation. The main technique in your paper about generating more language data has already been explored in 3D-LLMs in the scene context (e.g., Grounded 3D-LLM). Without clear novelty in the modeling or training methodology, reapplying or adapting those ideas to object-level tasks risks being seen more as an incremental extension than a fundamentally new contribution.
> >
> > I encourage the authors to more clearly articulate what is new in the method itself, and not just what is different in the task.

---

> ### Author Response · Authors · 2025-08-03
> **Explanation on Scene Level Tasks**
>
> The core focus of ShapeLLM-4o, as indicated by its name, is on object-level generation, editing, and understanding tasks. Scene understanding is neither the target of our research nor falls within the scope of our technical contributions. We argue that it is unreasonable and non-comparable for reviewers to request a comparison between ShapeLLM-4o and models specifically designed for scene understanding, for three main reasons:
>
> 1. **Analysis of Training Data Composition**: Object-level understanding datasets primarily consist of dialogue data focused on the object itself, emphasizing the capture of the object's geometric shape and detailed features. In contrast, scene-level task datasets focus on the overall scene, with the core focus being on the spatial relationships between objects. Although scene datasets can provide basic object descriptions, the granularity of these descriptions is insufficient for fine-grained tasks and cannot support detailed object-level question-answering tasks. Therefore, requesting the direct application of a model trained on object understanding datasets to be tested on scene-level datasets introduces a significant unfairness in evaluation. To validate this claim, we specifically tested the performance of Chat-Scene on object-level tasks—using the same experimental setup as PointLLM. The results showed a clear deficiency in its object understanding capabilities (see the table below for specific values). To ensure the fairness of the experiment, I sought confirmation from the original Chat-Scene author team through an anonymous channel, and they explicitly stated that the model's inability to transfer to object-level tasks was an expected outcome, with the underlying reason being the inherent differences in training data. This conclusion aligns perfectly with the core argument I made earlier. Existing research data suggests that similar works, such as PointLLM, ShapeLLM, and GPT4Point, focus on a single domain (scene or object) and have never been asked to balance both capabilities. Based on the above empirical analysis and consensus in the field, I must raise a clear objection to this evaluation method.
> || **B-1↑** | **R-L↑** | **M↑** | **S-BERT↑** | **S-CSE↑** |
> |:-----:|:------:|:----:|:-----:|:------:|:------:|
> | Chat-Scene|7.1|7.8|8.0|18.9|16.4|
>
> 2. **Analysis from a Technical Implementation Perspective**: Our model architecture and hyperparameter design are deeply optimized for object-level tasks. A typical example of this is the use of a 64-resolution voxel representation to balance fine-grained features with computational efficiency, along with the allocation of a 1024-length serialized encoding for each voxel. These design choices are primarily aimed at maximizing object performance. Directly transferring such specialized configurations to scene-level task testing would introduce evaluation bias due to the lack of domain adaptability.
>
> 3. **However, our model architecture exhibits outstanding transfer and scalability capabilities, requiring only minimal adjustments to seamlessly support scene-level tasks.** To validate this characteristic, we propose three simple and direct transfer strategies:
>
> **Strategy 1**: Independently encode each object in the scene into a 1024 sequence, and increase the inference batch size. After completing parallel captioning, the model output can be queried using the user-provided description.
>
> **Strategy 2**: Encode the scene as a voxel sequence and concatenate it with multiple views of the scene, directly inputting this into the model.
>
> **Strategy 3**: We can increase the compression rate of the current VQ-VAE, reducing the sequence length to 128 in order to accommodate all the objects in the scene. Additionally, we can assign the coordinates of each object's bounding box after each object.
>
> The three strategies outlined above will be explored in greater depth as part of our future work. It is important to emphasize once again that our model architecture has substantial value, enabling it to be seamlessly extended to scene understanding tasks.
>
> Understanding tasks represent just one dimension of this study's contributions. Our core innovation lies in the unprecedented unification of four major task frameworks: text-to-3D generation, image-to-3D generation, 3D object understanding, and 3D object editing. After in-depth discussions with the reviewers, we also plan to integrate 3D scene understanding into our framework in the future. We kindly request that the reviewers fully acknowledge the systematic breakthroughs our research brings to the field of cross-modal 3D generation and understanding, while understanding that scene understanding is not the core focus of this paper.
>
> Finally, we appreciate the reviewers' feedback, and we will incorporate this discussion into the main body of the text. Additionally, we will include relevant works (Chat-Scene, Grounded 3D-LLM, Robin3D, Chat-3D) in the related work section.

---

> > ### Comment · Reviewer_LdZP · 2025-08-03
> > **Problem solved; Please complete the Related Work**
> >
> > Dear Authors,
> >
> > Thank you for elaborating the differences between object-level and scene-level tasks, and why they should not be directly compared. This clarification is helpful.
> >
> > Please make sure to add the discussion on task differences and include works like Robin3D and Chat-3D in the related work section, as you mentioned. These additions will make the paper more complete.
> >
> > I'm willing to raise my score.
> >
> > Good Luck!

---

> > > ### Author Response · Authors · 2025-08-04
> > >
> > > Dear Reviewer,
> > >
> > > We are pleased that your concerns have been addressed, and we are very grateful for your revised score. Thank you for your thoughtful and constructive review again. We will include additional discussion and references in the final version of the paper.
> > >
> > > Good Luck!

---

### Official Review · Reviewer_pjk4 · 2025-07-03

**Clarity:** 4
**Significance:** 4
**Originality:** 3
**Rating:** 5
**Confidence:** 4

**Summary:**

This work demonstrates an effective approach to integrate 3D generation (text-to-3D and image-to-3D), understanding (3D captioning) and editing abilities into a pre-trained multimodal large language model (MLLM). In essence, they build a 3D foundation model. Furthermore, they introduce a large-scale 3D dataset, 3D-Alpaca, consisting of 2.56M data samples divided into 4 subsets, 1 for each of the following tasks: text-to-3D generation, image-to-3D, 3D captioning and 3D editing. Their approach involves first training a 3D VQVAE to encode voxel-based representations into compact discretized tokens which can be decoded back to voxels. They take Qwen2.5-VL-Instruct-7B as their backbone MLLM which is trained on their 3D-Alpaca dataset to extend its generative and understanding capabilities to 3D assets. The discretized token 3D representation facilitated by the 3D VQVAE enables them to feed these tokens as input to the model (for captioning and/or editing) and also to generate these tokens in an autoregressive next-token prediction paradigm, effectively treating them in a similar manner to discrete textual tokens. They demonstrate the efficacy of their model on all 4 tasks with extensive experiments to compare against previous baselines.

**Questions:**

1. Is the primary reason for using a voxel-based representation the ease of encoding given that voxels are inherently discrete? However, this comes at a cost such as loss in finer details which is why quite a few approaches have been explored to encode and/or compress continuous representations for any downstream task. Did you look into any of these representations and approaches? In essence, I just want to better understand the thinking that went behind the choice of using voxels.
2. It would help if some of the weaknesses mentioned above could be addressed.

**Ethical Concerns:**

["NO or VERY MINOR ethics concerns only"]

**Final Justification:**

This work presents an effective approach to extend a pre-trained MLLM with 3D generation, understanding, and editing capabilities, supported by the introduction of a large-scale dataset, 3D-Alpaca. The paper is clearly written, well-motivated, and demonstrates strong results across four tasks, with a simple yet scalable design that can integrate with various MLLMs. The rebuttal addressed all of my concerns sufficiently, including additional experiments on the VQ-VAE quality, image-to-3D evaluation, and limitations discussion. Thus, taking everything into consideration, I have raised my rating of this work.

**Limitations:**

Not sufficiently discussed.
1. Voxels have a limited resolution which means that they result in the loss of information and are not adequate to model finer details.
2. The data creation pipeline is dependent on several external models which means that the quality of the dataset is closely tied to the quality of the models used. While this approach to use existing SOTA models for synthetic dataset creation and/or labeling is becoming more common, specially given the rapid advancements made in some of these models, it is still important to acknowledge the inherent limitation on the quality of the dataset.

**Quality:**

3

**Strengths And Weaknesses:**

**Strengths:**
1. The paper is very well-written and was easy to understand.
2. This work addresses two important problems. There is a lack of 3D foundation models that can both understand and generate 3D assets and a shortage of adequate large-scale 3D datasets.
3. Use an additional text-based dataset, UltraChat, to ensure the model retains its original reasoning and dialogue capabilities.
4. Included many visuals to showcase the model's quality.
5. Simple but effective dataset curation approach.
6. Simply using discretized 3D tokens as input and output enables the seamless integration of 3D abilities into the model without significant changes to the core architecture. This further enhances the scalability of this approach as it can be applied to most MLLMs and is not strictly restricted to a specific model type.

**Weaknesses:**
1. It would have been to nice to see some results to evaluate the quality of their 3D VQ-VAE considering that it is a vital part of the methodology. Specifically, want to get an idea of how lossy the compression/encoding is. This can be checked by feeding some 3D assets from an unseen dataset (or even an unseen part of their own dataset but would be better if it is from a different dataset) to the VQ-VAE, getting the token representation, decoding it, and comparing the decoded asset with the original.
2. The comparisons against other models could have been a little more comprehensive. While it is not possible to compare against all relevant models, it would still have been better to cover some of the more popular models that have proven to be effective. For instance, for the 3D generative task, models such as OpenLRM, LGM, InstantMesh and Unique3D could have been included.
3. Not much discussion on the limitations of the work and the cons of some of the choices such as the downsides of using a voxel-based representation.

---

> ### Author Rebuttal · Authors · 2025-07-31
>
> We thank the reviewers for the constructive reviews. We provide our feedbacks as follows.
>
> > **Weaknesses1**
>
> **A1:** To assess the reconstruction quality of our 3D VQ-VAE model, we randomly select 1000 samples from the test set and feed them into the model. We then calculate several metrics between original and reconstructed voxel grids, including IoU, Recall, Precision, F1 and Chamfer Distance. These results, summarized in the table below, indicate that our 3D VQ-VAE model preserves geometric structure with high fidelity, providing a reliable reconstruction basis for following generation tasks.
> | **Method**   | **IOU**   | **Average Recall** | **Average F1** | **Average Precision** | **Chamfer Distance** |
> |:------------:|:---------:|:------------------:|:--------------:|:---------------------:|:--------------------:|
> | 3D VQVAE     | 0.9168    | 0.9357             | 0.9450         | 0.9549                | 0.0214               |
>
> > **Weaknesses2**
>
> **A2:** To provide a more comprehensive comparison, we quantitatively evaluate the image to 3D generation performance of OpenLRM, LGM, InstantMesh, and Unique3D on the same test set used in our paper. We also adopt the same evaluation metrics, including CLIPScore, FD_{incept}, and KD_{incept}. As shown in the table below, our method consistently outperforms all baselines and demonstrates superior 3D generation quality.
> | **Method** | **CLIP↑** | **FDincep↓** | **KDincep↓**(×100) |
> |:-------------:|:---------:|:------------:|:----------------------:|
> | Instantmesh   | 74.50     | 22.00        | 0.25                   |
> | OpenLRM       | 72.75     | 20.10        | 0.26                   |
> | Unique3D      | 77.10     | 16.80        | 0.13                   |
> | LGM           | 75.20     | 18.50        | 0.15                   |
> | ShapeLLM-4o   | **84.5**  | **12.2**     | **0.09**               |
>
> > **Weaknesses3**
>
> **A3:** Our current voxel-based representation provides effective structural alignment, enabling efficient tokenization with 3D VQVAE and seamless integration with autoregressive models. However, it does come with certain limitations. Specifically, voxel representations incur high memory and computational costs at higher spatial resolutions. Moreover, voxel-based representations are unable to model textures, and the use of low-resolution voxels limits ShapeLLM-4o's ability to understand 3D assets at a finer level of granularity. In future work, we aim to explore more efficient and finer-grained 3D representation methods.
>
> > **Questions1**
>
> **A4:** Thank you for your valuable suggestion. We will add some discussion on this section in the main text. In fact, we have considered other encoding methods. For meshtron, encoding mesh into discrete tokens through spatial quantization can result in excessively long sequences, which is not conducive to training autoregressive models. For continuous features represented by vecsets, we believe that on the one hand, their information content will be relatively rich, and it may be difficult to train a complete 3D vqvae for encoding. On the other hand, vecsets are implicit features, while voxels are explicit features, and 3D editing may require more explicit features. Considering downstream tasks, prevalent approaches (e.g., Trellis) increasingly adopt coarse-to-fine generation frameworks. Such pipelines typically first generate coarse voxel structures followed by high-resolution mesh refinement. Consequently, leveraging voxel representations enables ShapeLLM-4o to seamlessly integrate with contemporary 3D generation workflows, establishing greater generalizability.
>
> > **Limitations**
>
> To quantitatively assess the quality of our GPT-annotated text-to-3D dataset, we randomly sample 1000 text–image pairs and evaluate their semantic alignment using CLIPScore and ViLT R-Precision. As shown in the Table below, our dataset achieves high scores across both metrics, indicating strong correspondence between the captions and 3D content.
>
> |            | **CLIPScore** | **ViLT R-Precision R@5** | **ViLT R-Precision R@10** |
> |:----------:|:-------------:|:------------------------:|:-------------------------:|
> | **Ours**   | 29.58     | 35.3                 | 42.2                 |
>
> Similarly, to evaluate the alignment in 3D edting dataset, we compute the same metrics between the editing prompts and the rendered front-view of edited objects. The results, summarized in Table below, indicate the editing prompts are well aligned with the resulting modifications.
>
> |            | **CLIPScore** | **ViLT R-Precision R@5** | **ViLT R-Precision R@10** |
> |:----------:|:-------------:|:------------------------:|:-------------------------:|
> | Ours       | 27.41         | 33.6                     | 43.5                      |

---

> > ### Comment · Reviewer_pjk4 · 2025-08-04
> >
> > Thank you for sharing the additional experiments' results and the compelling explanations to my concerns. All my concerns have been sufficiently addressed and I am willing to raise my score. Just make sure to include the additional content, specially the VQVAE evaluation, the image-to-3D evaluation and the limitations discussion in the next revision of the paper (ideally in the main paper but supplementary is acceptable too if space is an issue). All the best!

---

> > > ### Author Response · Authors · 2025-08-04
> > > **thank**
> > >
> > > We are glad that your concerns have been resolved, and we sincerely thank you for your improved evaluation. Thank you again for your valuable and constructive comments. We will include the additional content you suggested in the final version of the paper. All the best!

---

### Note · Authors · 2025-08-12

We would like to sincerely thank the reviewers for their thoughtful feedback and valuable suggestions. Our ShapeLLM-4o introduces the first unified 3D-language model that supports 3D object generation (text-to-3D and image-to-3D), understanding (3D captioning), and editing within a single pre-trained multimodal large language model (MLLM). Additionally, we present 3D-Alpaca, a dataset that covers all necessary data for these tasks. To achieve unified representation, we train a 3D VQ-VAE to encode 3D meshes into discrete tokens. Our framework is built on Qwen2.5-VL-Instruct-7B, further trained on 3D-Alpaca to enhance its ability to generate and understand 3D assets. We evaluate our model across all four tasks, demonstrating strong results compared to previous baselines.

All reviewers (pjk4, LdZP, zpad, and Vmok) agree that our autoregressive framework represents a major step forward in unifying 3D object understanding and generation. They also highlight the value of our curated dataset, noting its large scale, diverse 3D asset types, and inclusion of both generative and understanding task data.

In our rebuttal, we addressed all reviewers' concerns. Regarding pjk4's concerns about the 3D VQ-VAE's reconstruction quality, we provide additional evaluations to confirm its accuracy. We also further explain our choice of voxel-based representation. For dataset quality concerns raised by pjk4 and LdZP, we present quantitative evaluation results for our GPT-annotated text-to-3D and 3D-editing datasets. Concerning LdZP's suggestion to evaluate on a scene-level benchmark, we clarify that our method is designed for object-level tasks, with direct comparisons not appropriate. We also propose potential directions for extending our framework to scene-level settings. Following requests from pjk4, zpad, and Vmok, we include more quantitative comparisons for 3D generation, understanding, and QA tasks. Lastly, regarding single-task performance compared to TRELLIS, raised by zpad and Vmok, we conducted a fine-tuning experiment and achieved comparable results on our test set.

We will incorporate all additional discussions, experiments, and related works mentioned in the rebuttal in the revised version of our ShapeLLM-4o paper. We are grateful for the reviewers' constructive feedback, which has greatly enhanced the quality and clarity of our work.

---

### Decision · Program_Chairs · 2025-09-17

**Decision:**

Accept (spotlight)

**Comment:**

This paper presents ShapeLLM-4o, a multimodal LLM for 3D generation and understanding. ShapeLLM-4o supports 3D object generation (text-to-3D and image-to-3D), understanding (3D captioning), and editing within a pre-trained model.  The paper reveals key components including the 3D VQ-VAE and a 3D-Alpaca dataset.

The reviewers acknowledged the novelty and contribution of this paper. After the rebuttal, the final ratings of the four reviewers are all positive: 1 accept and 3 borderline accept.

The AC agrees with the reviewers' ratings.
AC thinks this work advances the development of unified 3D foundation models with sufficient contribution, thus AC suggests accepting this paper with a spotlight.